# Distinct cortico-striatal compartments drive competition between adaptive and automatized behavior

**William H. Barnett**[1]*, **Alexey Kuznetsov**[2⊙]*, **Christopher C. Lapish**[1,3⊙]*

**1** Department of Psychology, Indiana University—Purdue University Indianapolis, Indianapolis, Indiana, United States of America, **2** Department of Mathematics, Indiana University—Purdue University Indianapolis, Indianapolis, Indiana, United States of America, **3** Stark Neurosciences Research Institute, Indiana University—Purdue University Indianapolis, Indianapolis, Indiana, United States of America

⊙ These authors contributed equally to this work.
* whbarnet@iu.edu (WHB); askuznet@iupui.edu (AK); clapish@iupui.edu (CCL)

**Data Availability Statement:** We have provided our source code at https://github.com/whbdupree/DMS_DLS_for_PLOSONE.

**Funding:** 5T32AA007462 to WHB from the National Institute on Alcohol Abuse and Alcoholism. https://www.niaaa.nih.gov/

## Abstract

Cortical and basal ganglia circuits play a crucial role in the formation of goal-directed and habitual behaviors. In this study, we investigate the cortico-striatal circuitry involved in learning and the role of this circuitry in the emergence of inflexible behaviors such as those observed in addiction. Specifically, we develop a computational model of cortico-striatal interactions that performs concurrent goal-directed and habit learning. The model accomplishes this by distinguishing learning processes in the dorsomedial striatum (DMS) that rely on reward prediction error signals as distinct from the dorsolateral striatum (DLS) where learning is supported by salience signals. These striatal subregions each operate on unique cortical input: the DMS receives input from the prefrontal cortex (PFC) which represents outcomes, and the DLS receives input from the premotor cortex which determines action selection. Following an initial learning of a two-alternative forced choice task, we subjected the model to reversal learning, reward devaluation, and learning a punished outcome. Behavior driven by stimulus-response associations in the DLS resisted goal-directed learning of new reward feedback rules despite devaluation or punishment, indicating the expression of habit. We repeated these simulations after the impairment of executive control, which was implemented as poor outcome representation in the PFC. The degraded executive control reduced the efficacy of goal-directed learning, and stimulus-response associations in the DLS were even more resistant to the learning of new reward feedback rules. In summary, this model describes how circuits of the dorsal striatum are dynamically engaged to control behavior and how the impairment of executive control by the PFC enhances inflexible behavior.

## Introduction

The repeated expression of a behavior is associated with the transition in the control of behavior from neural circuits that are optimized for flexible responding to those that are optimized

P60AA007611, R01AA029409, and INIAstress U24AA029970 to CCL from the National Institute on Alcohol Abuse and Alcoholism. https://www.niaaa.nih.gov/ This research was supported in part by Lilly Endowment, Inc., through its support for the Indiana University Pervasive Technology Institute. The funders had no role in study design, data collection and analysis, decision to publish, or preparation of the manuscript.

**Competing interests:** The authors have declared no competing interests exist.

for inflexible responding, and this transition involves neural circuitry in the basal ganglia that is devoted to habit and automaticity [1, 2]. Molecular changes resulting from chronic substance abuse alter the function of this circuitry from enabling the automaticity of routine tasks to driving compulsive drug seeking [3, 4]. Clearly articulating the computational processes that unfold across these circuits is critical for understanding the neural basis of behavioral control. Furthermore, it is critical to determine how these processes are altered in disorders that reflect the impairment of executive control, such as addiction.

The dorsomedial striatum (DMS) and dorsolateral striatum (DLS) are basal ganglia structures that have been implicated in the pathophysiology of inflexible behavior and drug addiction [2, 5]. In early training, drug seeking behavior is disrupted by inactivation of the DMS but is insensitive to inactivation of the DLS [6, 7]. A transition occurs after extensive training, where drug seeking is disrupted by inactivation of the DLS and is insensitive to inactivation of the DMS. While these data clearly outline distinct temporal roles of the medial and lateral compartments of the dorsal striatum in the control of behavior, how this transition occurs is unclear.

The distinct roles of the DMS and DLS in inflexible behavior are likely derived from differences in their computational properties. The DMS and DLS are, respectively, critical for goal-directed and habitual behavior [8, 9]. The inactivation of the DMS decreases sensitivity to reward devaluation [10], and lesions of the DLS increases sensitivity to reward devaluation and abolishes habitual seeking in the absence of a reward [11]. A potential mechanism for these differences was articulated by Yin et al. (2009) [12] that found plasticity in the DLS occurs relatively slowly compared to DMS as training progresses.

The differences in the computational functions of the DMS and DLS are also supported by differences in dopamine efflux and cortical inputs. There is evidence that different compartments of the striatum receive partition-specific nigro-striatal projections that encode different information [13, 14]. Classical RPE-encoding nigrostriatal neurons project to the DMS, and salience-encoding neurons project to the DLS [13]. Moreover, dopamine release in the basal ganglia acts on cortico-striatal synapses, and cortical input to the striatum is topographically organized across the DMS and DLS. Inputs to the DS exhibit a clear topographic bias where the medial portion of the striatum is more likely to receive input from the prefrontal cortex (PFC), and the lateral portion more likely to receive input from somatosensory and motor regions [15–17].

The input from PFC to the DMS is of particular interest. The PFC is important for cognition, executive function and goal-directed behavior [18–29]. Neuronal activity in the PFC contextualizes sensory inputs with respect to cognitive state and recent reward feedback [30–32], and these outcome representations are necessary for goal-directed learning [2]. Critically, these regions are impaired following prolonged alcohol use [33] and this impairment corresponds to increased responding for alcohol [34, 35]. In recent work, we investigated how the computational properties of the medial PFC are altered in a rodent model of excessive alcohol consumption and found impairments in neural signatures of the intent to drink and seeking behavior [36, 37]. Taken in combination with observations about the DMS and DLS, we leverage these results to illustrate a hypothesis for how impairment in the PFC contributes to the emergence of inflexible behavior.

Here we present a new computational model of cortico-striatal learning that incorporates goal-directed learning in the DMS and habit learning in the DLS. We derive this implementation from a theory of reinforcement learning based on dopamine-mediated plasticity of cortico-striatal projections to medium spiny neurons (MSNs) [1]. Dopamine induces long-term potentiation and long-term depression in D1 and D2 receptor-expressing MSNs respectively, and these changes are hypothesized to configure the basal ganglia to selectively disinhibit

**Table 1. Key additions that extend the model from prior works.**

| Experimental observation | Citation |
|---|---|
| DMS and DLS receive separate specific dopaminergic input | [13, 14] |
| DMS and DLS receive distinct cortical projections | [15–17] |
| Plasticity in the DLS is slower than plasticity in the DMS | [12] |
| Impairment of executive control is attributed to poor PFC coding for outcomes | [36, 37] |

thalamocortical relay neurons in the context of ongoing behavioral tasks [38]. This hypothesis has been incorporated in computational models that include reward-based learning that is based on RPE in the striatum [38–40]. Previous investigations have described behavioral controllers that delegate goal-based and habit learning to distinct model-based and model-free reinforcement learning modules [41–45]. Recent models of the basal ganglia have accounted for stimulus-response learning as a feature of cortical plasticity [39, 40, 46], as a feature of distinct dopaminergic reward coding in the DLS [47–49], or in the context of the spatial distribution of dopamine release in the medial-lateral axis of the dorsal striatum [50]. To understand the mechanism of how behaviors transition from goal-directed to inflexible, a computational model is required that is capable of articulating how changes in cortico-striatal plasticity support both goal-directed learning in the DMS and stimulus-response learning in the DLS. Based on four key assumptions (Table 1), we extend previous models that incorporate stimulus-response driven behavior to elucidate the roles of the DMS and the DLS in adaptable behavior and how the circuitry responsible for habitual responding could exhibit pathological primacy during inflexible behavior.

In the present study, we simulate a number of two-alternative forced choice behavioral tasks to investigate the interaction of the DMS and the DLS. Following an initial learning session, we challenge the model with reward devaluation, reward reversal, and a punished outcome (Fig 1). These different behavioral tasks are implemented by manipulations to the magnitude, action contingency, and valence of the reward. We challenge the model again in scenarios characterized by the impairment of executive control. In these simulations, neural activity of the PFC fails to appropriately code for action selection. Model performance is quantified by calculating the likelihood of action selection on a trial-by-trial basis over an ensemble of model instances. Our results demonstrate how impaired executive function could reduce the efficacy of goal-directed learning in the DMS and emphasize the expression of previously learned stimulus-response associations and habitual behavior by the DLS.

## Global structure of the model

### Organization of cortico-striatal partitions

In this study, we extend our previous model of the basal ganglia to implement a neural network that combines learning in the DMS and learning in the DLS to perform a two-alternative forced-choice decision-making task (Fig 2A). This new model implements choice-specific channels [38–40, 51–54] that compete to perform reward-based learning across partitions of the dorsal striatum [13, 14]. Choice-specific channels in this model correspond to representations of outcomes or actions in the neural activity of the cortex and basal ganglia. These channels are supported by the observation that MSN activity is organized into local ensembles of cells that encode outcome and action space [55].

The model can, broadly, be separated into two partitions: The medial partition incorporates the PFC and the DMS, and the lateral partition incorporates the pre-motor cortex (PMC) and

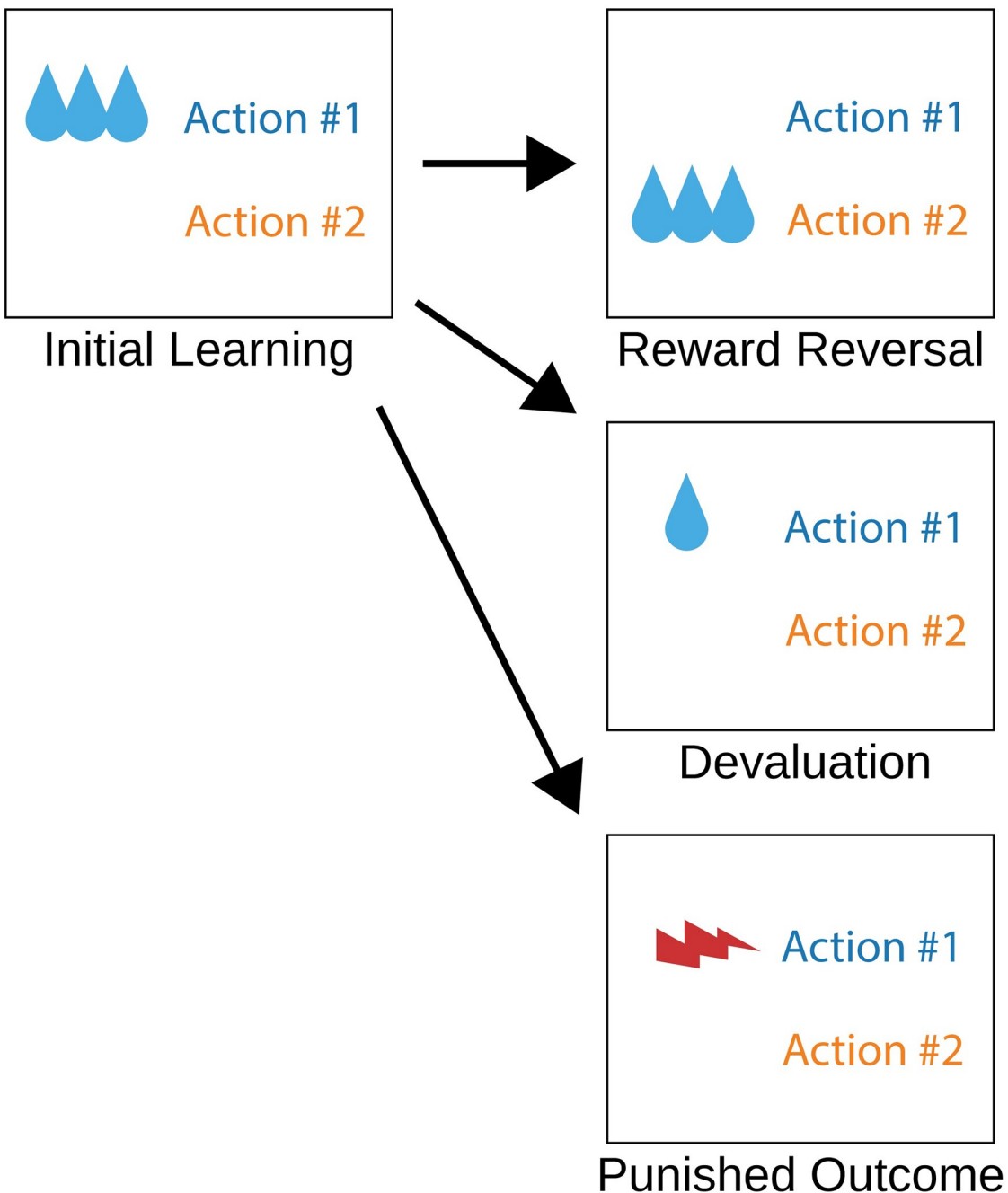

**Fig 1. Diagram of simulated behavioral sessions.** Model instances were trained with reward feedback that was specific to the behavioral session. The three other types of behavioral sessions each began after being trained by an initial learning session such that in the reward reversal, devaluation, and punished outcome sessions, the model instance was already trained to select action #1.

the DLS. Each partition also contains a direct and indirect pathway for each of the two behavioral choices that are presented to the agent. Each partition engages the machinery of the BG to utilize input from cortex, and the processing of this input modulates inhibition back onto the cortex in a way that reflects the accumulation of past experience in cortico-striatal weights. By implementing different synaptic weight update rules in the DMS and DLS, different modes

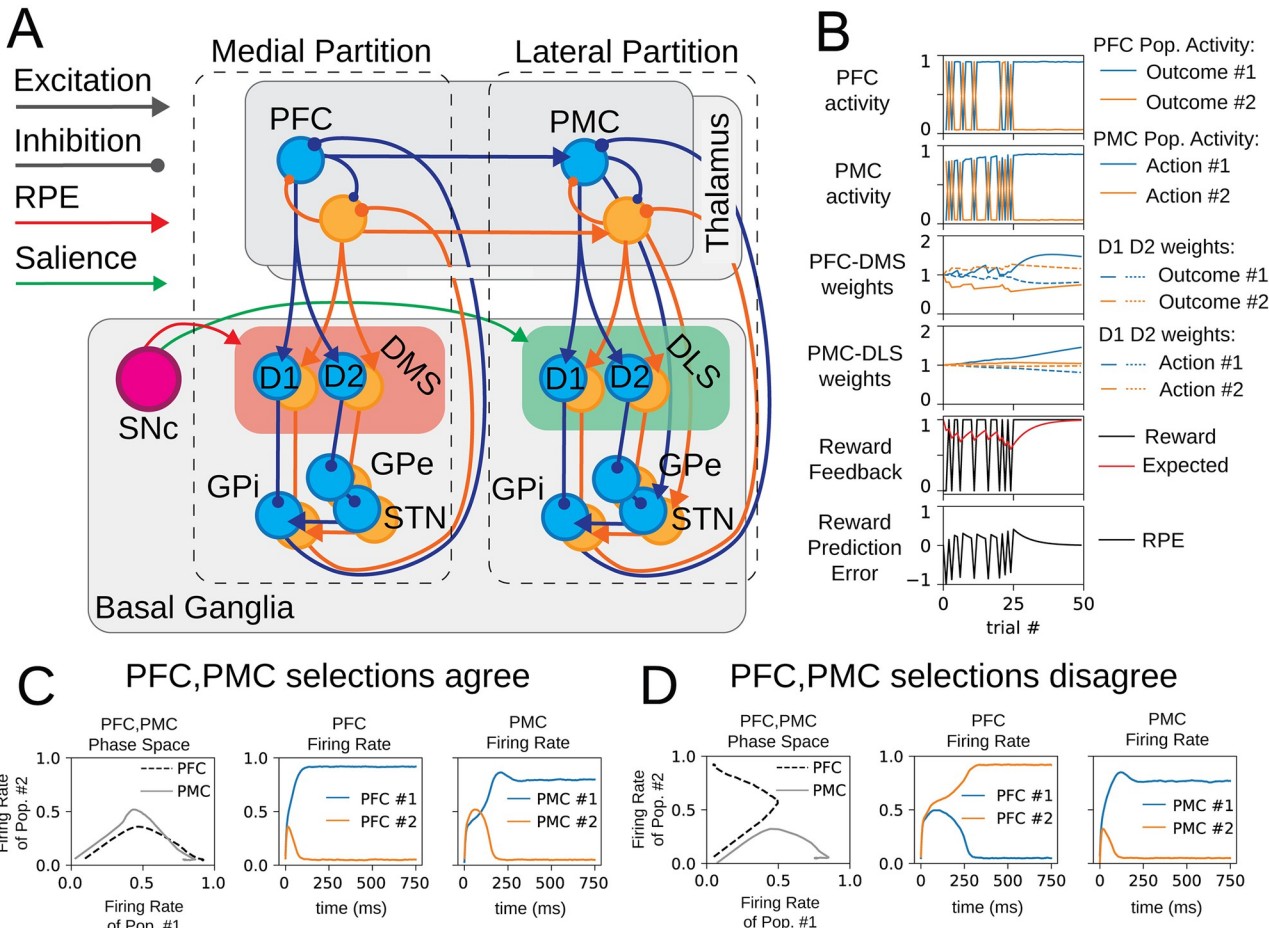

**Fig 2. A model of bimodal and concurrent learning in the basal ganglia.** (A) Cortico-basal ganglia-thalamo-cortical loops are distributed between the medial and lateral partitions of the model and are organized into distinct channels that represent individual actions (color-coded as blue and orange). The medial partition incorporates the prefrontal cortex (PFC) and the portion of the basal ganglia that includes the dorsomedial striatum (DMS). The lateral partition incorporates the premotor cortex (PMC) and the portion of the basal ganglia that includes the dorsolateral striatum (DLS). The DMS is coded red to indicate that dopamine release in this region codes for reward prediction error. The DLS is coded green to indicate that dopamine release in this region codes for salience. Within each compartment of the basal ganglia, each of the two channels processes cortical input via the striatonigral (direct) pathway and striatopallidal (indirect) pathway which converge on a combined node representing the substantia nigra pars reticulata and the globus pallidus internal (SNr/GPi). D1-expressing medium spiny neurons (MSNs) directly inhibit the SNr/GPi (direct pathway). D2-expressing MSNs project to the indirect pathway, which includes the globus pallidus external (GPe) and the subthalamic nucleus (STN). Within each partition, cortical excitation of MSNs is modulated by dopaminergic projections from the substantia nigra compacta (SNc). These dopamine signals encode different quantities within the DMS and the DLS and thus induce different modes of learning in each partition. In the DMS, dopamine modulates cortico-striatal synaptic weights based on the reward prediction error (RPE), and in the DLS, dopamine modulates cortico-striatal weights based on contextual salience. (B) Sample model activity during a behavioral session includes end-trial cortical activity, cortico-striatal synaptic weights, and reward feedback metrics. (C,D) Cortical partitions (PFC and PMC) perform independent outcome and action selection. PFC and PMC selection may agree (C) or disagree (D).

of learning are obtained over simulated successive forced-choice trials. These forced choices are evaluated in both the medial and lateral cortico-striatal loops. The neural activity in these two partitions disambiguates the neural representations of actions and the neural representations of outcomes that are contingent upon actions.

In this model, we distinguish between cortical inputs to the DMS and DLS, these inputs are the distinct cortical representations required to drive goal-directed learning and the formation of habit. The differentiation of cortical input to the DMS and DLS in this model is supported by observed cortical projections to the striatum [15–17]. Neural activity in the PFC is

conceptualized as outcome representations, and the selection of a particular outcome is indicated by the firing rate of corresponding PFC neural populations. Outcome selection is a binary variable determined by the firing rate that is highest between the competing populations in PFC, which indicates the most rewarding behavioral outcome. This outcome selection is informed by cortico-striatal weights in the DMS, which have been determined by reward prediction error (RPE) over successive trials (Fig 2B). Neural activity in the PMC corresponds to action representations, and the selection of a particular action is indicated by the firing rate of the corresponding PMC neural populations. Action selection is a binary variable determined by the relative firing rates of the PMC populations. This computation integrates outcome representations from the PFC as well as the results of accumulated stimulus-response association in DLS (Fig 2B). The selection of an action determines whether an agent receives rewarding (or punishing) feedback.

This network organization provides the structure for an outcome-to-action map. Since cortico-striatal weights in the DMS track reward prediction errors, the medial partition pairs the neural representations of outcomes with the reward continency of an action. In this way, the PFC-DMS drives behavior by computing desirable outcomes. On the other hand, the lateral partition (PMC-DLS cortico-striatal loops) retrieves what action is most likely to have been paired with the current stimulus. Feedforward coupling from PFC to PMC provides the association of outcome representations and action representations. This connectivity realizes an outcome-to-action map that learns to adapt to new reward feedback rules and receive reward in a consistent and predictable manner.

### Organization of the basal ganglia

The functional architecture of the BG in this model is derived from our previous models [39, 40], which are informed by Frank (2005). The BG is organized into two subsequent pathways that descend from the striatum: the direct pathway (striatum-SNr/GPi) and the indirect pathway (striatum-GPe-STN-SNr/GPi) (Fig 2A) [56–58]. The direct and indirect pathways converge on the SNr and the GPi, which are the output node of the BG [59–61]; here, these two regions are treated as the same populations. D1 MSNs directly inhibit the SNr/GPi. D2 MSNs inhibit the GPe which subsequently inhibits the SNr/GPi; in other words, D2 MSN activation leads to disinhibition of the SNr/GPi. Preferential activation of the direct pathway promotes the selection of an action. When D1 MSNs are active, they suppress SNr/GPi neurons that would otherwise inhibit the representation of that action in the cortex. On the other hand, preferential activation of the indirect pathway biases against the selection of that action. D2 MSN activation releases SNr/GPi from inhibition, and the corresponding cortical targets are suppressed. Learning in the BG is contingent on the release of dopamine (DA) from the SNc [62]; this reward-based signal induces long-term potentiation in D1 MSNs and long-term depression in D2 MSNs [63, 64]. By altering the ability of D1 and D2 MSNs to utilize cortical input, DA release supports reward-based learning in this system.

### A firing-rate model of the basal ganglia that includes both DMS and DLS

The computations that mediate the selection of the best of the two actions occurred concurrently within the medial and lateral partitions, and we distinguished the results of these two separate processes based on learning modality. Since learning in the medial partition was based on reward prediction error [13] (Table 1), the PFC-DMS weights were updated after each trial to accumulate information about reward contingency (Fig 2B). As such, we characterized learning in the medial partition as goal-directed learning, and we designated the choice made within the medial partition to be outcome selection. On the other hand, learning in the

lateral partition was based on salience of the stimuli to recently received reward. This signal contained information about recent reward history, and the update of PMC-DLS synaptic weights accumulated information about the likelihood that a given stimulus would result in a reward (Fig 2B). Learning in the lateral partition associated stimuli to rewarding responses. Since the output of the lateral partition defined behavior, we designated the choice made in the lateral partition to be action selection. In order to tune the strength of feedback that was appropriate for both medial and lateral partitions, we performed a number of simulations evaluating reward and punishment simulations for a range of strengths. We selected magnitudes of feedback for investigation based on consistent and robust response to reward and punishment (S3 Fig; see Methods).

Outcome selection was independent of action selection, and while these processes typically agreed they would occasionally disagree (Fig 2C and 2D). The firing rate of the PFC population determined outcome selection, and each PFC population excited the corresponding PMC population. In this way, outcome selection influenced action selection in the lateral partition. If the PFC population corresponding to outcome #1 was selected, the high firing rate of that PFC population made it more likely that action #1 was selected in the lateral partition, and in some cases both partitions agreed: outcome #1 and action #1 were both selected (Fig 2C). However, action selection in the lateral partition was also subject to its own learned stimulus-response associations, and the internal dynamics of the lateral partition competed with input from the medial partition to determine which action is performed. As a result, there were cases where the two partitions disagreed: PFC population #2 had a high firing rate indicating the selection of outcome #2, but in the lateral partition, PMC #1 had a high firing rate indicating the selection of action #1 (Fig 2D).

## Results

### Reward engages both goal-directed learning and the formation of a stimulus-response association

Learning in this model was accomplished by changes in the cortico-striatal synaptic weights. Generally, the enhancement of D1 synaptic weights promoted the selection of the corresponding behavioral choice, and the enhancement of D2 synaptic weights made the selection of the corresponding behavioral choice less likely. At the beginning of the initial learning session, D1 and D2 weights were not biased to select one of the two options (Fig 3A and 3B1 marker *1). Over many trials, the reward from the selected action was used to update these weights such that one action was more or less likely to be selected. As discussed below, the medial and lateral partitions utilized reward in different ways.

In the medial partition, the reward prediction error (RPE) was used to update cortico-striatal synaptic weights. This quantity could be either positive or negative and change sign from one trial to the next. For example, when there was a low expected reward and the rewarded action (in this case action #1) was chosen, the RPE became positive (Fig 3A and 3B2 marker *2). If the PFC population representing outcome #1 was active, the PFC-DMS D1 weight for outcome #1 was enhanced and the PFC-DMS D2 weight for outcome #1 was suppressed (Fig 3A and 3B3 marker *3). Since PFC population #2 was not active at this step, PFC-DMS weights for outcome #2 were not updated based on RPE; these weights slowly decayed to their baseline value.

On the other hand, if the unrewarded action (action #2) was chosen, the RPE became negative (Fig 3A and 3B4 marker *4). If the outcome #1 neurons were also activated in this case, the PFC-DMS D1 weight for outcome #1 was suppressed and the PFC-DMS D2 weight for outcome #1 was enhanced (Fig 3A and 3B4, 3B5 marker *5). However, if the unrewarded action

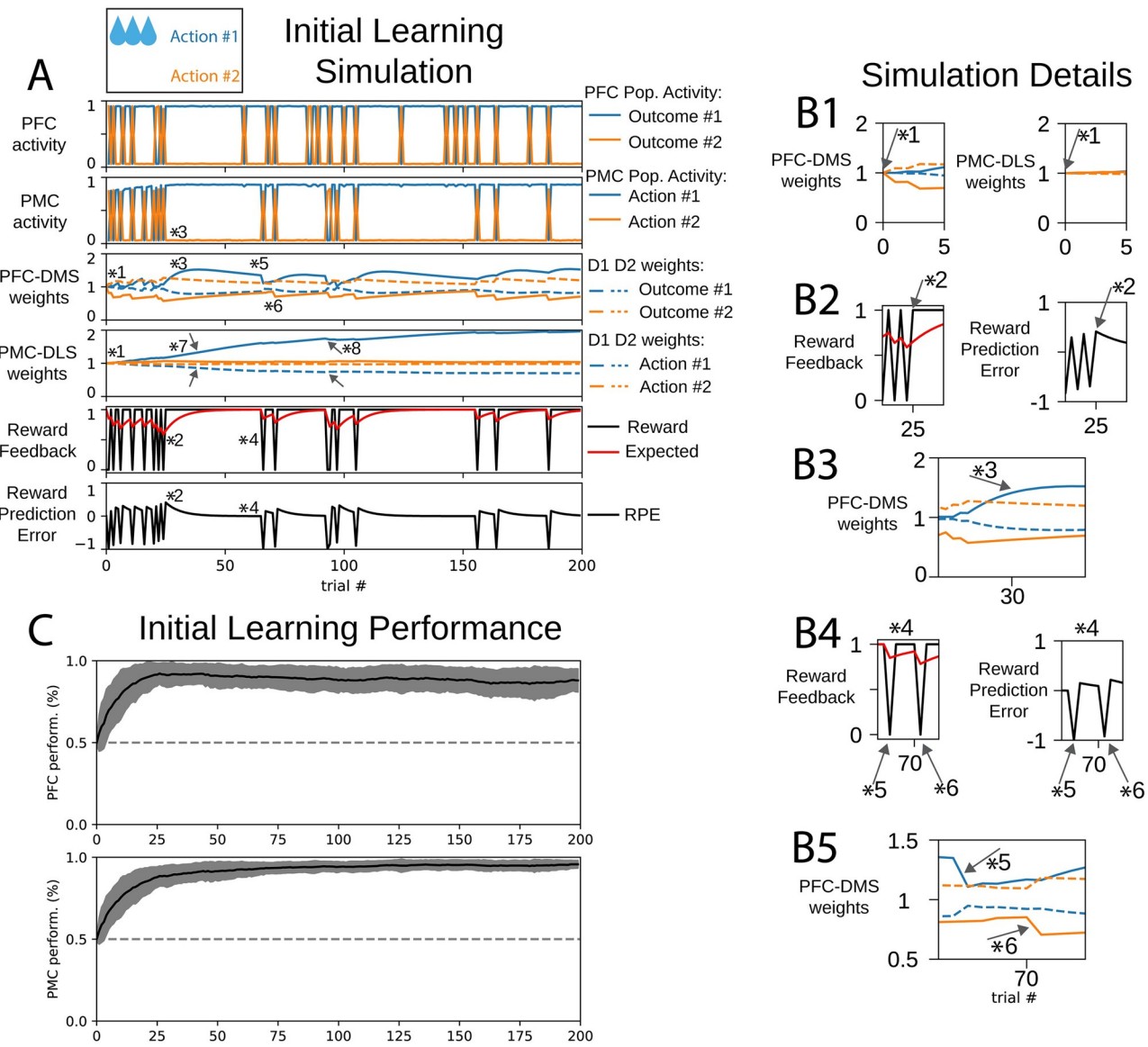

**Fig 3. An agent learns to select action #1 in a initial learning behavioral task.** (A) Cortical activity, cortico-striatal synaptic weights, and reward feedback metrics during initial learning behavioral session. In panels depicting cortico-striatal weights, D1 synaptic weights are solid traces and D2 synaptic weights are dashed traces. Blue traces correspond to action #1 and orange traces correspond to action #2. (B) Insets correspond to trials indicated by markers 1–6 in part (A). (B1) Initial D1, D2 weights were not biased to select either action. (B2) When the reward was greater than the expected reward, the reward prediction error (RPE) was positive. (B3) The PFC-DMS cortico-striatal synaptic weights evolve to promote the selection of outcome #1. (B4) When the expected reward is high and no reward is delivered, the reward prediction error becomes negative. (B5) Marker *5: selection of outcome #1 is de-emphasized when PFC selects outcome #1 and PMC selects the unrewarded action #2. Marker *6: selection of outcome #2 is de-emphasized when PFC selects outcome #2 and PMC selects the unrewarded action #2. (C) Learning performance of 100 agents. Performance is defined as the likelihood for an agent to selection action #1.

(action #2) is selected and the PFC selected outcome #2, the PFC-DMS weights were changed to de-emphasize the selection of outcome #2: the D1 weight was decreased, and the D2 weight was increased (Fig 3A, 3B4 and 3B5 marker *6). Over the course of training, the PFC-DMS weights were updated such that the PFC inputs to the DMS channel for outcome #1 efficiently activated the direct pathway, which emphasized the selection of outcome #1, and the PFC

inputs to the DMS channel for outcome #2 efficiently activated the indirect pathway, which tended to prevent the selection of outcome #2. Thus, the cortico-striatal weights in the DMS embodied a policy to promote outcome #1 and avoid outcome #2.

Recent data indicates that dopamine release in the DLS encodes behavioral salience rather than reward prediction error [13]. Here, we interpret salience to depend on past behavioral engagement and utilize expected reward to compute cortico-striatal weight updates in the lateral partition. In this session, expected reward was strictly positive. When the rewarded action (action #1) was chosen, the PMC-DLS D1 weight for that action was enhanced and the PMC-DLS D2 weight for that action was suppressed (Fig 3A marker *7). When the unrewarded action was chosen, the expected reward decreased but remained positive. The PMC-DLS D1 weight was also enhanced and its PMC-DLS D2 weight was also suppressed (Fig 3A marker *8). The differentiation of PMC-DLS D1 and PMC-DLS D2 weights could become biased across channels only when one action is selected many more times than the other action. Since the agent selected action #1 more frequently than action #2, the PMC-DLS D1 weight increased and PMC-DLS D2 weight decreased. In this way, the weights for action #1 ultimately became biased to promote the selection of action #1. Since action #2 was less frequently selected, the weight update between trials was dominated by the decay term, and these weights stayed near their baseline value. In this way, the DLS learns a stimulus-response association for the rewarded action but has accumulated no information about the unrewarded action. Behavior driven by this stimulus-response association is determined by the frequency at which this action was selected rather than contingency upon outcome.

## Reward reversal engages goal directed learning to promote a new stimulus-response association

Reward reversal simulations began with cortico-striatal synaptic weights inherited from the end of the initial learning session (Fig 1); this model instance was already biased to select action #1 (Fig 4A). In this session, the reward feedback was altered such that action #2 was the rewarded action. In early trials, the PMC continued to select the previously rewarded action #1 due to the persistence of PMC-DLS synaptic weights that favored action #1. The expected reward was high, and when action #1 was selected, the reward prediction error became negative. This selection suppressed the PFC-DMS D1 weight for outcome #1, enhanced the PFC-DMS D2 weight for outcome #1, and led to a decrease in the expected reward (Fig 4A and 4B1 marker *1). However, the PFC-DMS synaptic weights for outcome #2 required the selection of action #2 due to its dependence on the reward to be actively updated. Since the PMC-DLS weights were initially biased to select action #1, there was a low likelihood of selecting action #2 for a number of trials.

During early reversal trials, synaptic weights in the DMS became biased to favor the indirect pathway for both outcome #1 and outcome #2 (Fig 4A and 4B2 marker *2), and since the weights for outcome #1 and outcome #2 were very similar, there was not a strong likelihood for the DMS to emphasize one outcome over the other. The medial partition does not receive input from the lateral partition, so it is free to produce an outcome selection based on its own internal dynamics. Since synaptic weights in the DMS were biased to promote both indirect pathways, the PFC was equally likely to select either outcome #1 or outcome #2 for several trials. The similarity of weights for outcomes #1 and #2 encoded the agent's uncertainty about its environment, and by exploring outcomes, the PFC acted to learn the new reward contingency. Once the rewarded action #2 was selected, the corresponding PFC-DMS D1 synaptic weight was enhanced; the corresponding PFC-DMS D2 synaptic weight was suppressed; and the subsequent selection rate for outcome #2 increased in the medial partition (Fig 4A and 4B3

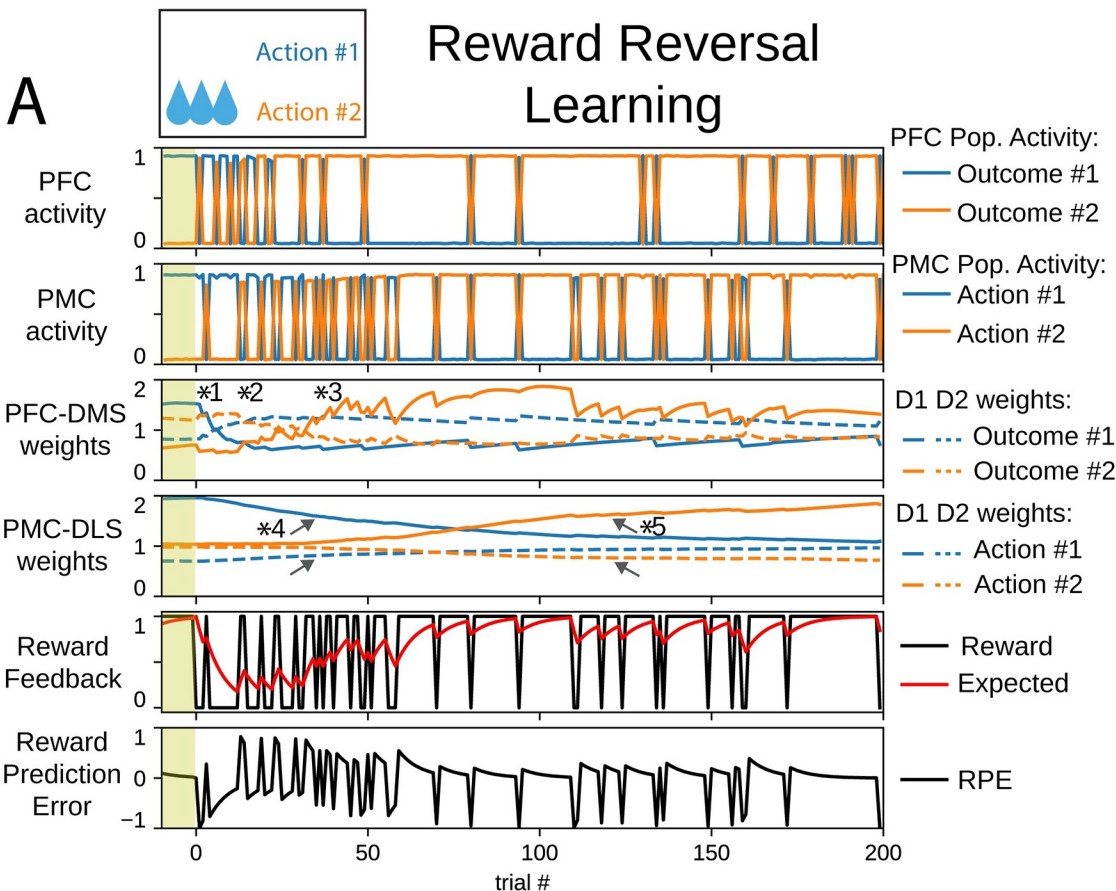

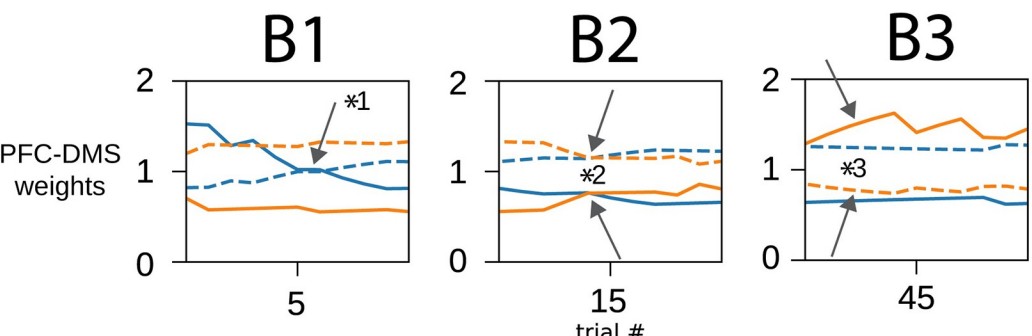

**Fig 4. An agent learns to select action #2 in a reward reversal behavioral task.** (A) In panels depicting cortico-striatal weights, D1 synaptic weights are solid traces and D2 synaptic weights are dashed traces. Blue traces correspond to action #1 and orange traces correspond to action #2. The yellow shaded trials before trial 0 indicate trials at the end of the initial learning session. (B) Insets correspond to trials indicated by markers in part (A). (B1) PFC-DMS cortico-striatal synaptic weights respond to the change in reward feedback rule to stop selecting outcome #1. (B2) The PFC-DMS weights for both channels emphasize the indirect pathway; this configuration promotes exploration in the PFC. (B3) PFC-DMS cortico-striatal synaptic weights evolve to promote the selection of outcome #2.

marker *3). The combination of occasional excitatory input to the PMC resulting from the selection of outcome #2 in the PFC and the gradual decay of PMC-DLS synaptic weights that favored action #1 lead to a transition in PMC action selection from the persistent selection of action #1 to exploration of action selection.

The decay of PMC-DLS synaptic weights towards the baseline is due to a decrease of the expected reward in early trials. The channel for action #1 in the DLS was less likely to promote action #1, and the decay of these weights contributed to exploration of actions in the lateral partition (Fig 4A marker *4). Concomitantly, goal-directed selection of outcome #2 in the PFC promoted the selection of action #2 in the PMC. During this exploratory phase, the occasional selection of action #2 increased the expected reward, and the PMC-DLS weights for action #2 began to differentiate—the PMC-DLS D1 weight for action #2 increased and the PMC-DLS D2 weight for action #2 decreased—such that the selection of action #2 was promoted (Fig 4A marker *5). In this way, the DLS lost the stimulus-response association for action #1 and learned a stimulus-response association for action #2. Over several trials, the decay of weights associated with action #1 allowed the emergence of exploratory action selection such that the alternative action #2 could be sampled and then subsequently exploited.

## Reward devaluation decreases selection accuracy

In the following simulations, we implemented reward devaluation by reducing the magnitude of the reward feedback in our two-alternative forced choice task. An agent was presented with the option to select the rewarded action #1 and the unrewarded action #2. Cortico-striatal weights were inherited from the end of the initial learning session during which the magnitude of reward for selecting action #1 was 1 (Fig 1); this model instance was already biased to select action #1. In this devaluation session, we decreased the reward value for action #1 to 20% of its value in the initial learning session. This is based on the observation that the dopamine response scales with the magnitude of the reward [65–67]. Animals that are sensitive to devaluation decrease their responding following the reduction in the value of a rewarding stimulus [68, 69]. Following extended training, animals may lose sensitivity to the value of a rewarding stimulus [68]. Here, we probe biophysical mechanisms featured in this model that support sensitivity to devaluation and the resistance to devaluation.

In this session, the expected reward began close to 1, however, the selection of action #1 provided decreased reward value. The expected reward was large compared to the reward received, and the RPE became negative in early trials despite the selection of the rewarded action (Fig 5A and 5B1 marker *1). Since the RPE was negative, PFC-DMS D1 weights for outcome #1 were suppressed and PFC-DMS D2 weights for outcome #1 were enhanced in early trials: the channel for outcome #1 in the medial partition was biased to select against outcome #1 (Fig 5A and 5B2 marker *2). However, the selection of action #2 was not rewarded; there was no change in the valence of PFC-DMS weights for outcome #2. The agent was less likely to select outcome #1, but it was still more likely to select outcome #1 over outcome #2. As it integrated the newly reduced reward value, the expected reward converged to a new value; in these trials, action selection pushed the expected reward close to 0.2, and mistakes in action selection reduced the expected reward below 0.2. Once the expected reward followed this new regime, the exploratory phase in outcome selection was abolished. The RPE became positive again when the rewarded action was selected, and the PFC-DMS weights for outcome #1 were restored to select for outcome #1 (Fig 5A and 5B3 marker *3).

The change in magnitude of the reward altered the steady state value of PMC-DLS weights. Since the reward was diminished, the expected reward was also relatively small. The expected reward modulates the magnitude of change in PMC-DLS synaptic weights, so the amount by

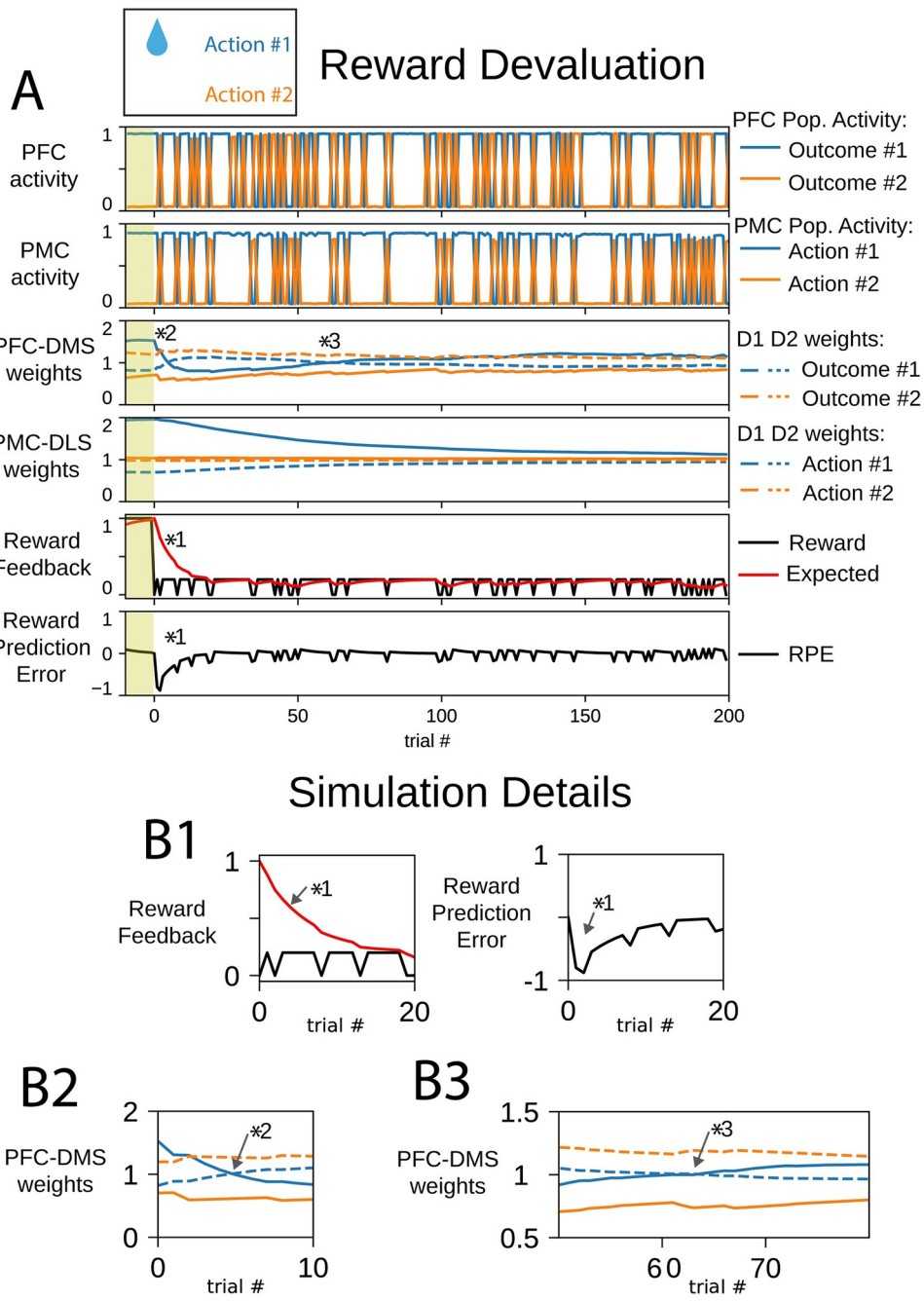

**Fig 5. The magnitude of reward feedback is reduced in a reward devaluation behavioral task.** The selection of action #1 persists but becomes less likely. (A) In panels depicting cortico-striatal weights, D1 synaptic weights are solid traces and D2 synaptic weights are dashed traces. Blue traces correspond to action #1 and orange traces correspond to action #2. The yellow shaded trials before trial 0 indicate trials at the end of the initial learning session. (B) Insets correspond to trials indicated by markers 1–3 in part (A). (B1) Following reward devaluation, the reward feedback is less than the expected reward. Even though, reward feedback continues to be mostly positive, the reward prediction error (RPE) is negative. (B2) The cortico-striatal synaptic weights for outcome #1 reflect the negative reward prediction error, and the indirect pathways for both outcomes are emphasized. (B3) Since action #1 is rewarded, the cortico-striatal synaptic weights for outcome #1 eventually recover to emphasize the direct pathway over the indirect pathway.

which action selection changed PMC-DLS weights was relatively small. This active update of synaptic weights was required to prevent decay of weights to their baseline values, and over many trials, the PMC-DLS D1 and D2 weights converged to new values that were closer to baseline synaptic weights. While the stimulus-response association for action #1 was diminished, it was not lost entirely. Following reward devaluation, the PMC-DLS D1 and D2 weights decreased towards baseline but were still configured to promote the selection of action #1. The agent persistently select action #1 with reduced reliability. These results also suggest that the formation of habit is sensitive to the magnitude of the reinforcing stimulus.

## Punishment engages goal-directed learning

We altered the reward for action #1 to be negative. The selection of action #1 resulted in punishing or aversive feedback, and the selection of action #2 was unrewarded (or unpunished). This punished outcome session began with cortico-striatal weights inherited from the end of the initial learning session (Fig 1); this model instance was already biased to select action #1. In this session, the magnitude of the reward for action #1 was 50% that in the initial learning session, but it had the opposite valence. We define this negative reward to be punishment, which is based on experiments showing that pauses in DA firing are the dominant response to aversive stimuli [13, 70]. Rodents readily learn to avoid aversive stimuli [71–73]. In these simulations, we examine how punishing feedback could engage the disparate learning mechanisms of the dorsal striatum.

In early trials of the punished outcome session, the PMC continued to select the previously rewarded action #1 due to the persistence of PMC-DLS synaptic weights that favored action #1 (Fig 6A). In the first trial, the expected reward was close to 1. Selection of either action #1 or action #2 decreased the expected reward since the reward received was never positive. Similar to the early trials of the reward devaluation session, the RPE was strongly and consistently negative at the beginning of this session. During these trials, the PFC-DMS D1 and D2 weights for outcome #1 changed valence such that the DMS channel for outcome #1 emphasized the indirect pathway over the direct pathway, and the PFC performed outcome exploration (Fig 6A and 6B1 marker *1). During exploration, the PFC-DMS weights for both channels emphasized the indirect pathway (Fig 6A and 6B2 marker *2). Finally, PFC-DMS weights for outcome #2 began to emphasize the direct pathway and suppress the indirect pathway as the PFC began to persistently select outcome #2 (Fig 6A and 6B3 marker *3). The progression of PFC-DMS weights in this session was similar to the progression of weights in the reward reversal session.

In these simulations, we altered the feedback used by the DLS to update cortico-striatal synaptic weights. Substantia nigra dopaminergic neurons that project to the DLS respond to both rewarding and aversive stimuli with a positive dopamine transient [13, 14]. During punishment sessions, the lateral partition utilized the *rectified* reward expectation to perform weight updates. The rectified expectation integrates the absolute value of the reward. This signal is intended to reflect the behavioral salience of the selection option. In all previous sessions, the reward was positive and the rectified expectation was identical to expected reward. However, in the punished outcome sessions, the rectified expectation diverged from the expected reward.

The combination of decreased selection of action #1 and the reduced rectified expectation prevented the agent from maintaining PMC-DLS D1 and D2 weights for action #1 and deemphasized this action. As these weights converged to their baseline values, the PMC selected the unpunished action #2 more preferentially than punished action #1 based on input from the PFC. However, the PMC-DLS weights for action #2 did not strongly develop to select for

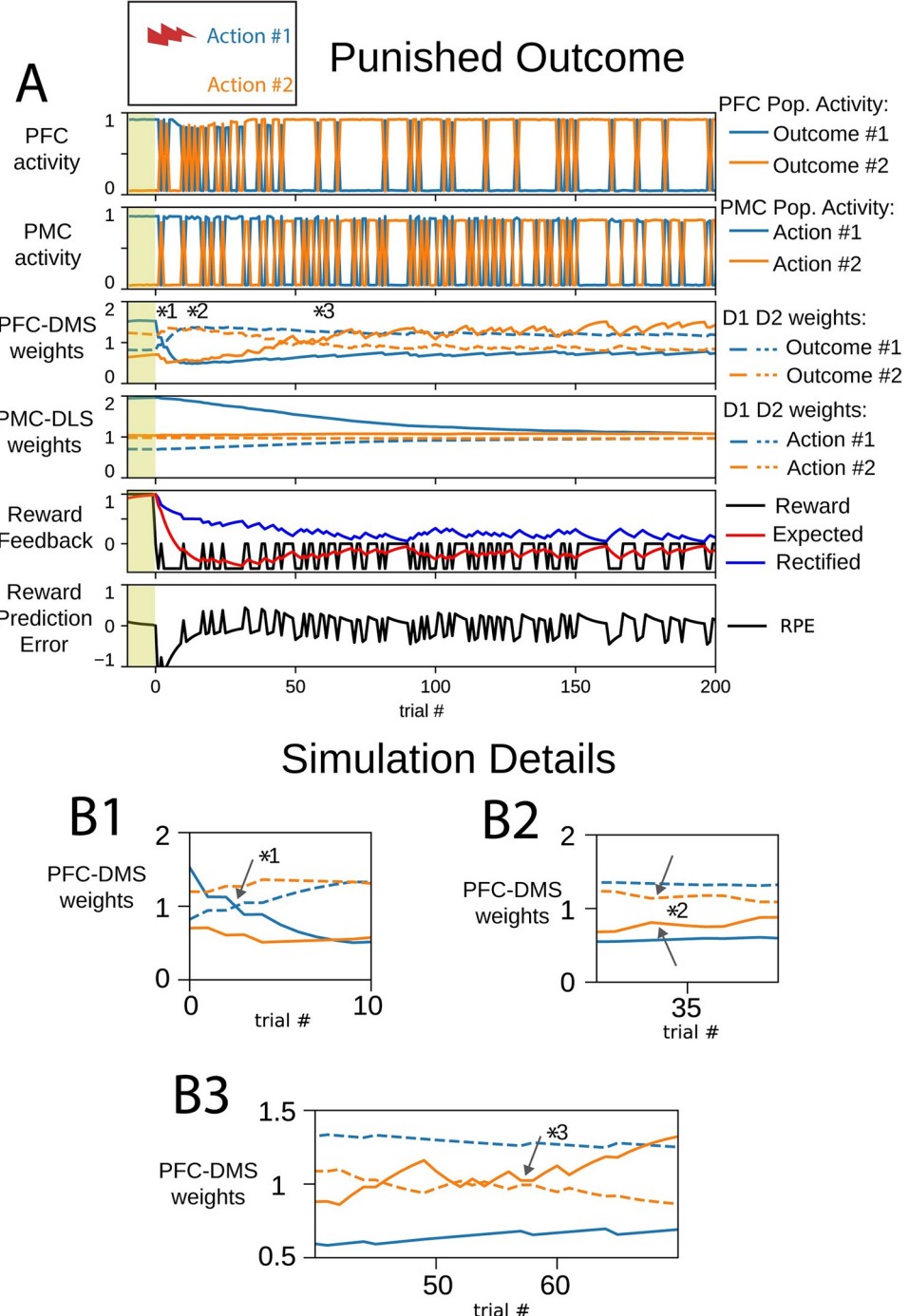

**Fig 6. An agent learns to select action #2 in a punished outcome behavioral task.** Cortical activity, cortico-striatal synaptic weights, and reward feedback metrics evolve to overcome habit to select action #1. The yellow shaded trials before trial 0 indicate trials at the end of the initial learning session. (A) In panels depicting cortico-striatal weights, D1 synaptic weights are solid traces and D2 synaptic weights are dashed traces. Blue traces correspond to action #1 and orange traces correspond to action #2. (B) Insets correspond to trials indicated by markers 1–3 in part (A). (B1) PFC-DMS cortico-striatal synaptic weights for outcome #1 respond to the negative reward prediction error and evolve to emphasize the indirect pathway. (B2) During exploration, the PFC-DMS channels for both outcomes emphasize the indirect pathway. (B3) In the channel for outcome #2, the direct pathway becomes emphasized to promote the selection of the unpunished choice.

action #2, and action selection was driven by the PFC which sought to minimize the delivery of punishment. This operation minimized the rectified expectation (kept it close to zero). Since the synaptic plasticity update rule for PMC-DLS depends on rectified expectation, which had a small value, the change in PMC-DLS synaptic weights was dominated by the decay of synaptic weights after each trial. Therefore, learning to avoid a punished outcome does not develop the stimulus-response association of choosing an alternative action.

In simulations where the magnitude of punishment was large, model instances could adopt a maladaptive behavior, which was characterized by repeated selection of the punished action after learning has already occurred (S4 Fig). In these cases, the rectified expectation became large very quickly when the punished action was selected several times in a row by happenstance. The large rectified expectation in these cases facilitated the stimulus-response association for the punished action, and the model instance would go on punished runs. These runs were terminated by goal-directed learning and subsequent changes in the in the PFC-DLS connections, where selection of unpunished outcome discouraged the selection of the punished action.

## Misalignment of PFC outcome selection with channels in DMS and PMC decreases action selection performance

Work from our group shows that outcome representations are reduced in animals that drink alcohol compulsively, and also demonstrate impaired behavioral flexibility [36, 37, 74]. More specifically, reductions in mutual information about two competing options was observed in neural recordings obtained from the mPFC of rats that drink alcohol compulsively, indicating that the representation of these options was not clearly disambiguated. To simulate this observation, we repeated reward reversal and punished outcome simulations incorporating reduced outcome coding in the PFC (Fig 7). In these simulations, each PFC population routed 90% of its projections to the appropriate channel and 10% of its projections to the wrong channel (striatal neurons corresponding to the other outcome) (Fig 7A). The inhibitory input from the GPi populations to the PFC was similarly mis-matched to retain balance between excitation and inhibition in these networks.

These changes in connectivity in the model are intended to reflect a change in the fidelity of prefrontal-cortical representations. The output of each PFC channel to the DMS and the PMC, and the feedback to the PFC from the GPi is contaminated by information from the opposing channel. In this way this impairment lead to a PFC outcome signal that is slightly contaminated by the alternative outcome. We interpret this implementation of impaired executive control as a disruption of the outcome-to-action mapping. The mechanisms for goal-directed learning in the DMS operate on representations that do not cleanly map to representations in the PFC, and the PFC projections into the PMC do not cleanly promote either action #1 or action #2. In simulations of reward reversal (S1 Fig and Fig 7B) and punished outcome (S2 Fig and Fig 7C), impaired outcome coding in the PFC delayed the learning of the new reward feedback rule. We examined performance of outcome-selection and action selection for a group of 100 model instances. In both sessions, performance curves for outcome selection in the PFC and action selection in the PMC were shifted up and to the right indicating a reduction in performance following impairment of outcome coding in the PFC.

To characterize the change in agent performance due to impaired outcome coding in the PFC, we performed a change point analysis on outcomes and actions. In each model instance, change points in cortical activity were detected as the first zero-crossing of the log likelihood ratio in a normative model of evidence accumulation. This analysis was applied separately to PFC and PMC activity (Fig 8A) and yielded a trial number for the change point

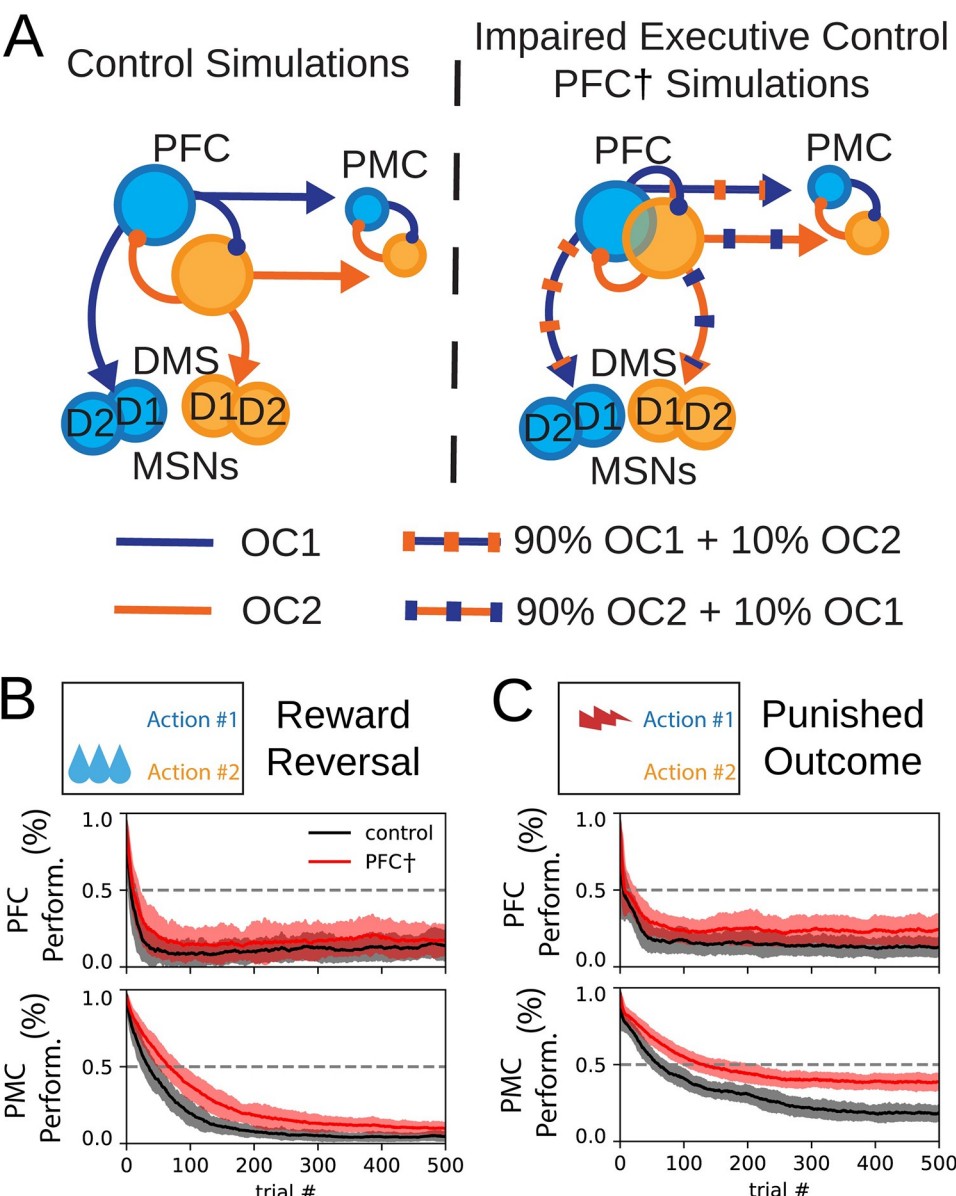

**Fig 7. Impairment of PFC decreases learning performance.** (A). Illustration of impairment of PFC coding. The projections of the PFC into the DMS and the PMC contained mixed signals in PFC† behavioral sessions. OC1 and OC2 correspond to projections that represent the channels for outcome #1 and outcome #2. (B-C) Performance is defined as the likelihood for agents (N = 100) to selection action #1; here, agents learn to select action #2. (B) Progression of outcome selection in the PFC and action selection in the PMC during a reward reversal task (black) and a reward reversal task with impaired PFC (red). (C) Progression of outcome selection in the PFC and action selection in the PMC during a punished outcome task (black) and a punishment learning task with impaired PFC (red).

of both cortical compartments. For each behavioral session, we performed this analysis for 100 distinct agents (see Methods). We compared outcome selection and action selection change points between simulations of reward reversal and reward reversal with the impaired outcome coding in the PFC (PFC†) and between punished outcome and punished outcome PFC† sessions (Fig 8B). From the reward reversal to the reward reversal PFC† sessions, the median agent change point significantly increased from trial 15 to 21 in outcome in the PFC

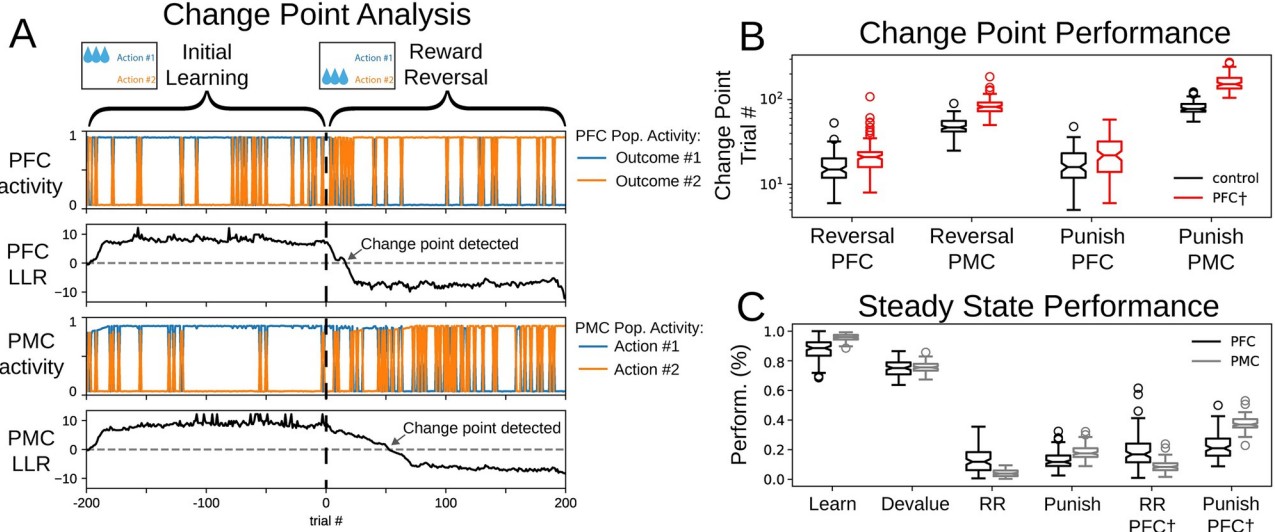

**Fig 8. Analysis of model output across behavioral tasks.** (A) Example model activity of an initial learning session and the following reward reversal session with change point analysis. Trial number here is relative to the beginning of the reward reversal session. (B) Change point analysis indicates the trial at which an ideal observer detects a change in cortical selection from action #1 to action #2. We compared change point performance in control simulations (black) and sessions with impaired PFC (red) (N = 100 agents). Box and whisker plots with median depicted. (C) Performance of agents at the end of each behavioral task showing the likelihood that agents select action #1. We compared PFC performance (black) to PMC performance (grey) (N = 100 agents). Box and whisker plots with median depicted.

(Mann-Whitney U = 2441.5, $n_1 = n_2 = 100$, p = 3.90E-10) and from trial 47.5 to 85 in the PMC (Mann-Whitney U = 134.5, $n_1 = n_2 = 100$, p = 1.35E-32). From punished outcome to the punished outcome PFC[†] sessions, the median agent change point significantly increased from trial 16 to 22 in the PFC (Mann-Whitney U = 3705.5, $n_1 = n_2 = 100$, p = 1.56E-3) and from trial 81 to 162.5 in the PMC (Mann-Whitney U = 87.5, $n_1 = n_2 = 100$, p = 3.41E-33). Thus, the simulations bridge the compromised outcome coding in the PFC and reduced performance in the action selection tasks.

## Discrepancy between steady state performance of goal-directed and stimulus-response learning

In several of the behavioral challenges, steady state agent performance differed between the PFC in the medial partition and the PMC in the lateral partition. Steady state performance was defined as the mean agent performance at the end of each behavioral session (Fig 8C). In the initial learning session and in reward reversal sessions, steady state performance was greater in the PMC than the PFC; the PMC was more likely to choose the rewarded action than the PFC was to choose the corresponding outcome (Table 2). The better performance in the PMC reflects the accumulated contribution of the past choices encoded in the PMC-DLS weights, which indicates a learned stimulus-response association. In the devaluation session, there was no significant difference between PFC and PMC performance (Table 2). In punished outcome sessions, the PMC was less likely to choose the unpunished action than the PFC was to choose the corresponding outcome (Table 2). We also observed a significant difference in steady state performance of action selection between initial learning and devaluation sessions (Mann-Whitney U = 10000, $n_1 = n_2 = 100$, p = 2.2E-16).

**Table 2. Discrepancy between goal-directed learning and habit.**

| Session Name | Mann-Whitney U statistic | p-value |
|---|---|---|
| Initial learning | 1958.5 | 1.1E-13 |
| Devaluation | 4070 | 0.023 |
| Reward reversal | 8333.5 | 3.8E-16 |
| Punished outcome | 1737.5 | 1.6E-15 |
| Reward Reversal PFC[†] | 8352.5 | 2.6E-16 |
| Punished outcome PFC[†] | 907.0 | 1.5E-23 |

In each test, $n_1 = n_2 = 100$. Comparisons can be visualized in the notched box-and-whisker plots on Fig 8C.

## Discussion

The main aim of this study was to formalize the computational mechanisms that support the expression of goal-directed and habitual behavior, as well as the transitions between them. Here, we showed how a stimulus-response association—embodied in the accumulation of cortico-striatal synaptic weights in the DLS—could prevent agents from quickly learning new reward feedback rules during simulated behavioral tasks. Indeed, the selection of the previously rewarded action perseverated even after cortico-striatal weights in the DMS were configured in a manner to support exploration. Specifically, DLS-driven action selection countered goal-directed behavior despite reward reversal, reward devaluation, and the punishment of an outcome, and in these scenarios, we characterized behavior as habitual.

In this work, we utilized the anatomical and physiological segregation of dopamine signals in the striatum to demonstrate computational mechanisms that drive separate processes for flexibly tracking changes in the way that outcomes are contingent on environmental cues (goal-directed learning) and becoming adept at responding to familiar stimuli with well-practiced action (habitual learning). Our work follows recent investigations that approach similar problems exploring how behavior is influenced by different learning strategies employed in the brain. The control of goal-directed and habitual behavior have been attributed to concurrent model-based and model-free learning mechanisms by which an agent predicts the values of available actions [41, 42]. In model-based learning, an agent evaluates the consequences of actions across a decision-tree of possible outcomes [42]. This search for solutions requires an agent to evaluate outcomes for actions not yet taken or putative errors for actions not performed [75–77]. These approaches have been used to describe behaviorally-inspired controllers that manage the two learning strategies based on their comparative advantages [43–45]. These models frame control of learning strategy as competition between the prefrontal and dorsolateral striatal systems [41] or competition between the hippocampus and dorsolateral striatum [78]. Miller et al. [48] have recently challenged previous models for habit in reinforcement learning and propose a habit mechanism to describe stimulus-response associations based on Thorndike's second Law of Exercise—behaviors that have been repeated often in the past are more likely to be performed in the future. Their mechanism for value-free learning is similar to the action of the DLS presented in the present report, though we have implemented our model as neural circuitry. The learning mechanisms described in [48] emphasize goal-directed behavior under variable-ratio reinforcement schedules and stimulus-response behavior under variable-interval reinforcement schedules, which will be critical to address in future iterations of our model. Bogacz [47] has developed a mathematical framework to describe goal-directed learning and habit as a process based on Bayesian inference. Their algorithm is broken down into basic units that are compared to the anatomy and physiology of the basal

ganglia including separate components for the DMS and DLS. This framework differs from our own model by including circuitry that estimates and controls for uncertainty to weight the output of the goal-directed and habit systems [47]. Baladron and Hamker [46] have implemented simulations of the neural circuitry in the DMS and DLS respectively as goal-directed and stimulus-response controllers. Their modeling framework differs from our own by storing the synaptic weights responsible for habitual behavior in the infralimbic cortex, and these weights are slowly updated based on a dopamine-independent Hebbian plasticity learning rule [46]. We extend this body of work by demonstrating how a lack of fidelity in neural representations at the level of the PFC could facilitate the dominance of habitual responding. We accomplish this result by implementing an impairment of outcome representations in the PFC, which is dynamically involved in learning in our model of cortico-striatal interactions. Here, we implemented a simple circuit-level manipulation in our computational model to illustrate the emergence of inflexible behavior through primacy of the DLS in action selection. The expression of choices that were previously rewarded in the face of changing contingency is a hallmark of inflexible behavior, and the DLS is critical for the expression of inflexible behavior [2, 6]. On the other hand, the DMS is critical for goal-directed behavior, and disruptions to the dorsal striatum that favor the DMS may restore goal-directed behavior in the face of drug seeking [6, 7]. In compulsive drinking, PFC neural firing rates code poorly for outcome selection [36, 37]. We developed a prediction for inflexible behavior based solely on PFC impairment. If the representations of outcomes in the PFC fail to map cleanly to actions, then goal-directed learning in the DMS will lose efficacy, and agents will take longer to overcome previously learned stimulus-response associations. We implemented this impairment of executive control by slightly contaminating outcome representations with their alternatives in devaluation, reward reversal, and punished outcome tasks. Agents with impaired PFC took longer to learn new reward feedback rules and exhibited worse steady state performance.

## Reward devaluation and punishment reveal mechanisms that support habitual behavior

Experimental protocols may induce reward devaluation through manipulations that remove drive, for example by selective satiety, or induce an aversive association with the reward, for example using lithium chloride [68]. In our model design, we distinguish between reward devaluation tasks—which we accomplish by directly reducing the quantitative value of the reward for action #1, and punished outcome tasks—in which the valence of the feedback associated with action #1 changes from positive to negative. Both of these protocols provide the opportunity to evaluate the role of learning in the DMS and the DLS to overcome an established stimulus-response association. Through direct inspection of the models neural activity and cortico-striatal synaptic weights, we have shown, during the trials immediately following a change in outcome contingency, how stimulus-response associations drive behavior in spite of rapid goal-directed learning and updated reward-outcome contingencies in the PFC neural representations. These dynamics are an illustration of a mechanism by which neuronal circuitry that is critical for habitual responding resists or overpowers the action of circuitry that is critical for goal-directed behavior, and we classify the inflexible behavior driven by these dynamics as habit.

Punishment simulations were motivated in part by experiments in rodents that included a foot shock stimulus [13]. The reward-based feedback communicated via DA to the DLS—the rectified expectation—integrates the absolute value of the trial reward and is distinct from the expected reward. In this way, the dopaminergic signal to the DLS is positive for either a rewarding or an aversive stimulus (as in [13]). In goal-directed behavior, the aversive choice is

avoided; in this case the agent seeks to minimize the negative reward prediction error and, hence, the quantity of negative expected reward. Since the rectified expectation is minimized by the learning process, no new stimulus-response association is able to develop in punished outcome tasks. The results of learning in punished outcome tasks differ from the results of reward reversal. In both cases, the stimulus-response association for a previously rewarded action must be overcome. In the reward reversal session, goal-directed behavior works to maximize the reward delivered for the newly rewarded action. Hence, the expected reward is maintained at a large value over the course of training, and a new stimulus-response association is formed. This critical difference between reward reversal and punished outcome tasks—whether there is a new stimulus-response association or not—is visualized in the comparison of steady state performance (Fig 8B). In reward reversal simulations, the performance of outcome selection in the PFC is not as great as action selection in the PMC (discussed below). Thus, steady state behavior is driven by the new stimulus-response association formed in the DLS. However, in punished outcome tasks, steady state behavior is maintained by goal-directed learning (the medial cortico-striatal partition) in the absence of a new stimulus-response association in the DLS. The performance of outcome selection in the PFC leads performance of action selection in the PMC (Fig 8B). These results lead to the hypothesis that the control of behavior cannot be acquired by DLS, thereby coming under habitual control, in tasks where punishment is used as a reinforcer, and therefore habits will not form. These results can also be examined in the context our reward devaluation sessions. Here, the reward magnitude is a fraction of that during initial learning. Despite this change, the small reinforcer is still sufficient to engage learning in the DLS. Even though responding is less reliable in reward devaluation sessions than initial learning sessions, responding resists dramatic changes D1 and D2 weights of the DMS during early trials that would otherwise indicate reversal and selection of the alternate outcome (Fig 5A and 5B).

In the present work, the magnitude and valence of reward (or punishment) are fixed as a part of the definition of the simulation session. The perceived value of reward feedback may be modulated by experience. In rodents, following devaluation, the response of an animal to a rewarding stimulus may recover to pre-devaluation levels [79]. In humans, the avoidance of punishment becomes a rewarding rather than neutral experience [80]. The present model compares the expected reward to the reward received, and this comparison may yield a positive reward prediction error when, for example, an agent avoids the delivery of a punishing stimulus. However, the perceived magnitude of the reward is not encoded by a neural population in the model. A mechanism to adapt the perception of reward feedback according to context and experience could allow a similar model to lose sensitivity to the magnitude of reward feedback with experience or to develop a strong stimulus response association to the avoidance of a punishing stimulus. In this way, it is thought that the DLS is responsible for habits of avoidance, but these circuits are not as well studied in the context of active avoidance as in appetitive learning behavior [71–73]. Experimental evidence implicates the DLS in the later stages of active avoidance learning [81, 82], but it is not clear if the DMS and DLS play the same role in active avoidance as in appetitive learning [83].

## PFC coding failure exacerbates inflexible behavior

Our implementation of impaired executive control is intended to represent a failure of outcome representations in the PFC to appropriately map to action representations in the PMC. The prefrontal-cortical input to channels for outcome #1 and outcome #2 in the DMS and the PMC populations for action #1 and action #2 were slightly mixed in PFC[†] sessions. Specifically, the input to the DMS channel for outcome #1 and the input to the PMC population for

action #1 was a weighted some of both PFC populations: 90% PFC for outcome #1 + 10% PFC for outcome #2 (Fig 7A). The firing rates of these populations at the end of the simulation were used to update cortico-striatal synaptic weights in the DMS between trials of the session (see Methods). During selection of an outcome in the PFC, the PFC population corresponding to the selected outcome had a high firing rate, and the PFC population corresponding to the unselected outcome had a low firing rate (Fig 2C and 2D). Due to the manipulation, the firing rate used to update synaptic weights for the selected outcome is slightly lesser compared to control simulations, resulting in a slightly lesser change in synaptic weights. On the other hand, the firing rate used to update synaptic weights for the unselected outcome is slightly greater compared to control simulations, resulting in an inappropriately large change in the corresponding synaptic weights. Similarly, the excitatory input from PFC to PMC was less distinct between channels: the PFC was slightly less able to promote the selection of the appropriate action in the PMC in PFC[†] sessions compared to control simulations.

Our simulations of PFC impairment are intended to illustrate how impaired executive control in the PFC could promote the emergence of inflexible behavior. The PFC is critical for adaptive behavioral control, and impairment of the PFC disrupts cognitive flexibility [84]. Our results show a mechanism by which the impairment of the PFC could reduce goal-directed responding and facilitate the emergence of inflexible behavior. After loss of executive control, agents have reduced performance in simulations with altered action-outcome contingencies, and these differences appear in both the outcome representations in the PFC and in action-selection performance as indicated by PMC (Fig 8B). This deficiency in cognitive flexibility can be related to behavioral performance of animals with PFC dysfunction. Deficits in reversal learning performance have been associated with lesions or alcohol-induced damage to the orbitofrontal cortex [85–87].

How could repeated exposure to alcohol change the brain to induce a failure for neural activity to properly code for action selection? Dopamine dynamics in the PFC are critical for higher functions and executive control [88, 89], and changes to dopamine modulation in the PFC are concurrent with alcohol-induced deficits in executive function [90]. Patients that suffer from a history of alcohol abuse may have deficient expression of D2 dopamine receptors in the PFC [91, 92]. In rats, chronic exposure to alcohol disrupts D2 receptor function in the PFC [93]. It is thought that D1 receptors in the PFC act to stabilize network activity and D2 receptors promote flexibility [94, 95]. Disruptions in the PFC dopamine system that favor D1 receptors may render an individual unable to quickly adapt in the face of changing behavioral context.

## The role and organization of direct and indirect pathways

In reversal learning and punishment follow-up sessions, we observe two prominent phases in the evolution of cortico-striatal weights in the DMS. First, the BG channel for the previously rewarded action becomes essentially inactivated. The cortico-striatal weights for D1 MSNs decreased in value below baseline, and the weights for D2 MSNs increased above baseline. This transition was called out in Figs 4B1 and 6B1. During this phase, the model performance changed to become less likely to select the previously rewarded action, and as a result, it explored alternative actions—here there was only one alternative action. Second, this exploration phase was terminated by the potentiation of the cortico-striatal weight for D1 and depression of D2 MSNs that correspond to the newly rewarded action. A prediction that emerges from these results is that depotentiation of D1 MSNs and potentiation of D2 MSNs are critical for the transition to exploratory behavior. This prediction is supported by recent results

showing that after initial learning the behavioral flexibility is reduced in a task that requires learning new action-outcome associations by optogenetic inhibition of indirect pathway MSNs [96].

The observation that both direct and indirect pathway MSNs are activated at the onset of movement supports the hypothesis that direct and indirect pathways are concurrently involved in selection decisions [97–99]. In the "complementary" model of the striatum, clusters of direct pathway MSNs are selectively activated while indirect pathway MSNs broadly inhibit competing clusters [100]. In the present model, each action is represented by a cluster of direct pathway MSNs within the striatum. However, the present model differs from the "complementary" model in that each cluster of direct pathway MSNs acts in opposition to a selective cluster of indirect pathway MSNs. The computation performed in the SNr/GPi output node of our model is based on competition between the direct and indirect pathways associated with each action; in other words, the BG channel for each action emits a GO or a NOGO signal. In this way, our computational model is most like the "competitive" model described in [100] where direct and indirect pathway MSNs compete within an ensemble to promote or inhibit an action. Complex interactions between subtypes of MSNs challenge the rigid separation of direct and indirect pathways [97]. Moreover, dopamine action within the GP, STN or SNr may functionally contribute to animal behavior [101]. Our model, which operates at the level of action representations, uses multiple concurrent channels to represent distinct action options, and the firing rate of neuronal populations within these channels determines signal processing within the BG. This model is intended to capture the functional computation involved in action selection. For simplicity, the plasticity of synaptic weights is implemented at the level of the striatum to modulate the relative strength of the direct and indirect pathway and competition between actions is implemented via mutual inhibition within each cortical compartment.

## Striatal dopamine distribution

The spatio-temporal distribution of dopamine release within the striatum is critical for selective reward credit assignment [50]. However, dopamine release supports different learning modalities in different compartments of the striatum [13, 14]. Moreover, dopamine release occurs in wave-like patterns that propagate in the medial-lateral axis of the dorsal striatum, and the direction of wave propagation depends on behavioral context [50]. Specifically, tasks in which reward depended on animal performance promoted dopamine release beginning in the dorsomedial striatum, and tasks in which reward was independent of animal behavior promoted dopamine release beginning in the dorsolateral striatum. The authors [50] performed computer simulations to illustrate the hypothesis that spatiotemporal gradients in dopamine release could emphasize reward credit assignment to specialized striatal subregions and recruit behavioral strategies appropriate for the task at hand. Temporal coincidence in the activity of multiple signals may be involved in functional segregation of the striatal subregions [102, 103]. A joint consideration of both spatio-temporal [50] and modally [13, 14] mixed dopamine signals could further differentiate the function of different compartments of the striatum.

## Predictions and limitations

A key feature of learning is the transition from exploration to persistently performing a rewarding behavior. Here, we show via simulations how different learning modalities across the DMS and DLS could interact to contribute to this transition. The balance between exploration of unknown options and exploitation of known rewards is a ubiquitous challenge

necessary for the survival of animals [104]. Even after learning new reward-feedback rules, animals sample alternate actions, and this alternative sampling is representative of continual exploration [105, 106]. The neurophysiological basis for exploration and strategic control is an active topic of research with focus in the cortex [107, 108].

Our results here suggest a mechanism for the involvement of the DMS in the support of alternative sampling after a reward contingency rule has been learned. Once behavior becomes driven primarily by stimulus-response association—in other words by the DLS in the lateral partition—cortico-striatal synaptic weights in the DMS are not required to maintain agent behavioral performance. What role, then does outcome selection in the medial partition play after extensive training? At steady state, we observed a higher error rate in the outcome representations of the PFC than in the action selection as indicated by PMC (Fig 8C). When action selection was very accurate and driven by PMC-DLS weights, there were a paucity of strongly corrective events in the RPE signal compared to early learning. As such, the RPE-dependent term in the PFC-DMS weight update rule remained relatively small for more trials and the decay term of the PFC-DMS weight update rule determined steady-state weight values closer to baseline. Hence, the goal-directed learning unit became more likely to make mistakes. In other words, the PMC-DLS drove performance action selection up, but mistakes or lapses in the medial partition increase the likelihood of alternative sampling in action selection. We formulate this observation as a key prediction of our computational model: neuronal activity in the PFC should occasionally and increasingly correspond to alternative unrewarded actions as an animal learns to persistently select a rewarded action. In this way, the PFC and DMS could monitor alternative reward contingencies and contribute to tonic exploration.

Drug use or risk of drug use is associated with impairments in cognitive flexibility [109–113]. Following chronic exposure to drugs, animals are more likely to make perseverative errors immediately following a change in reward rule [105, 114]. We relate these results to our reward reversal PFC† simulations: in simulations with impaired outcome coding in the PFC, agents exhibit a substantial increase in perseverative errors (Fig 8A). Since model behavior in early trials was dependent on goal-directed learning, the impairment of outcome coding in the PFC compromised the ability of agents to overcome stimulus-response associations for previously rewarded actions.

The participation of the cortical inputs directly to the STN—the hyperdirect pathway—are proposed to participate in action selection by broadly inhibiting cortical units associated with the motor programs involved in action selection [115–117]. This inhibition precedes activation of the direct pathway and, hence, the specific disinhibition of the selected motor program in the thalamus and cortex. In a previous model, we investigated the role of the hyperdirect pathway in the emergence of BG oscillatory activity in Parkinson's disease [40]. In the current study, our implementation of the hyperdirect pathway is relegated to the lateral partition. The activation of the hyperdirect pathway is directly involved in selection of actions in the PMC rather than indirectly by way of inhibition of outcome representations in the PFC. In this way, the hyperdict pathway can facilitate top-down interruption of motor programs in behaviors such as the stop-signal task. This implementation is simplistic and a shortcoming of this model. However, we do not simulate behavioral experiments, such as the stop-signal task, which designed to probe the architecture of additional cortical inputs to the BG. [118, 119]. It will be critical to more adequately address the role of this pathway in future work.

Our model implements the cortex in a limited fashion, and in future work, this aspect of the model could be extended to capture greater anatomical and physiological detail. In our previous models, stimulus-response associations were captured by Hebbian learning in

cortico-cortical synaptic weights [39, 40]. Plasticity in these projections was omitted in the present study to emphasize the role of DLS in the acquisition of stimulus-response associations and habit. However, cortical mechanisms for learning are critical for cognitive and executive flexibility [120]. The inclusion of cortical learning would permit contingency representations external to the striatum and allow us to investigate the cortical dopamine dynamics of learning tasks in healthy animals and in animals that suffer from the chronic use of addictive substances. Additionally, our present model does not distinguish between different regions of the prefrontal cortex. The prelimbic and infralimbic cortices appear to play opposing roles in learning [2, 121]. Disruption of infralimbic cortex prevents the expression of habit and facilitates goal-directed behavior [122, 123], and the interaction of the prelimbic and infralimbic regions is critical for the performance of set shifting tasks [124]. Expanding this model to include the specific functions of these brain regions may accommodate aspects of behavior where learning resembles an immediate state change (e.g. attentional set shifting) as opposed to gradual changes over trials. The present model lacks an explicit implementation of the thalamus. As in Kim et al. [39] and Mulcahy et al. [40], we do not attribute dynamics in the thalamus to the action selection process. For simplicity, we have collapsed the basal ganglia-thalamus-cortical paths into inhibitory projections from the output node of the basal ganglia to cortical populations.

## Methods

### Neuronal simulations

Neuronal populations are modeled by their firing rates, and these firing rates are defined by ordinary differential equations. The biophysical parameter values of these differential equations were drawn from the previous published instance of this model [40] and then tuned by the authors to facilitate the new model architecture and simulations in this study. Key changes to the previous model were motivated by critical experimental constraints described in the Introduction (see Table 1).

The instantaneous firing rate $A$ is modeled as

$$\tau \frac{dA}{dt} = \sigma(I) - A + N(t)$$

The time constant is set to 12.8 msec for the STN, 20 msec for the Gpe, and 15 ms for all other neuronal populations [125]. The properties of neuronal populations are determined by the connectivity of neural circuitry, which was manifest as differences in their input currents ($I$). The activation function was

$$\sigma(I) = \begin{cases} 0, & if\ I \leq 0 \\ \tanh(I), & if\ I > 0 \end{cases}$$

The term $N(t)$ is a noisy process with uniform distribution and amplitude 0.1. Numerical integration is performed using the Euler method with a timestep of 0.15 ms.

The neuronal connectivity is distributed within and across network partitions. Here, the indicator $DXS$ represents one of either the dorsomedial striatum ($DMS$) or the dorsolateral striatum ($DLS$), and the indicator $CTX$ represents one of either the prefrontal cortex ($PFC$) or the premotor cortex ($PMC$). Partitions are organized such that the $DMS$ receives selective input from the $PFC$ in the medial partition and the $DLS$ receives selective input from the $PMC$ in the lateral partition. With the exception of the STN, the definition of BG populations is identical between the medial and lateral partitions. Circuitry within the BG is organized to

participate in cortico-BG-thalamo-cortical loops which compete via inhibition at the level of the cortex. These loops correspond to individual processing channels within the BG. Within the each of the medial and lateral partition, there are two competing loops that correspond to action #1 and action #2.

$$I_{D1} = g_{CTX} w_{CTX-D1} CTX$$

$$I_{D2} = g_{CTX} w_{CTX-D2} CTX$$

$$I_{GPe} = dr_{GPe} - w_{D2-GPe} D2 + w_{STN-GPe} STN$$

$$I_{STN} = I_{STN,DXS}$$

$$I_{GPi} = dr_{GPi} - w_{D1-GPi} D1 - w_{STN-GPi} STN$$

The variable for each neuronal population corresponds to its firing rate. Input currents were determined by the weights of their synaptic inputs ($w$)—which modified the firing rate of the presynaptic population, their excitatory drives ($dr$), and in some cases a gain (g). The input current of the STN differs between *DMS* and *DLS* partitions. The STN populations within the DLS receive hyperdirect pathway input from the PMC:

$$I_{STN,DMS} = dr_{STN} - w_{GPe-STN} GPe$$

$$I_{STN,DLS} = dr_{STN} - w_{GPe-STN} GPe + w_{HD} PMC$$

Mutual inhibition within the cortex (PFC-to-PFC inhibition and PMC-to-PMC inhibition) is tuned to force winner-takes-all choices. This outcome selection within the PFC is influenced by inhibitory inputs from the $GPi_{DMS}$. Action selection within in PMC is influenced by excitatory input from the PFC as well as inhibitory input from the $GPi_{DLS}$.

$$I_{PFC_m} = dr_{PFC_m} - w_{GPi-PFC} GPi_{DMS} - w_{PFC_n-PFC_m} PFC_n$$

$$I_{PMC_m} = dr_{PMC_m} + w_{PFC-PMC} PFC_m - w_{GPi-PMC} GPi_{DLS} - w_{PFM_n-PMC_m} PMC_n$$

## Sessions with impaired outcome coding in the PFC

In PFC[†] sessions, we alter the connectivity to and from the PFC in order to simulate scenarios in which the representations of outcomes in the PFC do not cleanly map to representations of actions in the PMC. A pair of effective PFC (*eff*PFC) firing rates were produced, which is a weighted average of the firing rate of the two PFC populations:

$$eff\,PFC_m = 0.9\,PFC_m + 0.1\,PFC_n$$

Notice that the "wrong" channel $PFC_n$ contributes 10% to *eff*PFC$_m$. If each channel contributed equally (50%) to *eff*PFC$_m$, then the PFC would be completely uncorrelated to action selection: *eff*PFC$_m$ and *eff*PFC$_n$ would each contain the same signal. Since the contribution 10% is one fifth of the way to equal contribution to *eff*PFC, the PFC is said to be misaligned with action selection by 20%. In these sessions, *eff*PFC firing rates are used in place of PFC firing rates to compute the synaptic currents on D1 MSNs, D2 MSNs, and in PMC populations. Similarly, the inhibitory input onto PFC populations from the *GPi* are

**Table 3. Model parameter values.**

| Parameter name | Parameter value |
|---|---|
| $g_{PFC}$ | 0.4 |
| $g_{PMC}$ | 1 |
| $dr_{GPe}$ | 1.6 |
| $w_{D2-GPe}$ | 2 |
| $w_{STN-GPe}$ | 0.4 |
| $dr_{STN}$ | 0.8 |
| $w_{GPe-STN}$ | 1 |
| $w_{HD}$ | 0.3 |
| $dr_{GPi}$ | 0.2 |
| $w_{D1-GPi}$ | 1.4 |
| $w_{STN-GPi}$ | 1.6 |
| $dr_{PFC}$ | 1.5 |
| $w_{GPi-PFC}$ | 1.8 |
| $w_{PFC-PFC}$ | 1.6 |
| $w_{PFC-PMC}$ | 0.1 |
| $w_{GPi-PMC}$ | 1.8 |
| $w_{PMC-PMC}$ | 1.6 |
| $\lambda_{DMS,D1}$ | 0.05 |
| $\lambda_{DMS,D2}$ | 0.025 |
| $\lambda_{DLS,D1}$ | 0.0025 |
| $\lambda_{DLS,D2}$ | 0.00125 |
| $d$ | 0.02 |
| $w_0$ | 1 |

adjusted to be a combination of inputs from competing loops:

$$I_{PFC_m} = dr_{PFC_m} - w_{GPi-PFC}\left(0.9\,GPi_{DMS,m} + 0.1\,GPi_{DMS,n}\right) - w_{PFC_n-PFC_m}PFC_n$$

In this way, PFC activity is adjusted to carry less information about action selection.
Model parameters are defined in Table 3.

## Synaptic plasticity

Dopaminergic input to the DMS codes the reward prediction error, which is the difference between reward ($R$) and the expected reward (*ExpRew*).

$$SNc_{DMS} = R - ExpRew_j$$

$$ExpRew_{j+1} = \alpha R_j + (1 - \alpha)ExpRew_j$$

The constant $\alpha$ takes value 0.15, and *ExpRew*$_j$ refers to the expected reward of the j*th* trial. The coefficient $\alpha$ determines how new reward is weighted against previous expected reward to determine the next value for expected reward. For most simulations, the dopaminergic input to the DLS is identical to the expected reward. However, when the reward feedback is negative (indicating an aversive stimulus), the absolute value of the reward is used to compute the

rectified expectation (*RectExp*).

$$SNc_{DLS} = RectExp_j$$

$$RectExp_{j+1} = \alpha |R_j| + (1 - \alpha)RectExp_j$$

The quantities $SNc_{DMS}$ and $SNc_{DLS}$ respectively represent the dopaminergic input to the DMS and DLS, which are utilized after each trial to updated cortico-striatal weights. Incremental updates to these weights are determined by the following expressions:

$$\Delta w_{PFC-D1,DMS,m} = \lambda_{DMS,D1} * SNc_{DMS} * PFC_m * D1_{DMS,m} - d * \left(w_{PFC-D1,DMS,m} - w_0\right)$$

$$\Delta w_{PFC-D2,DMS,m} = -\lambda_{DMS,D2} * SNc_{DMS} * PFC_m * D2_{DMS,m} - d * \left(w_{PFC-D1,DMS,m} - w_0\right)$$

$$\Delta w_{PMC-D1,DLS,m} = \lambda_{DLS,D1} * SNc_{DLS} * PMC_m * D1_{DLS,m} - d * \left(w_{PMC-D1,DLS,m} - w_0\right)$$

$$\Delta w_{PMC-D2,DLS,m} = -\lambda_{DLS,D2} * SNc_{DLS} * PMC_m * D2_{DLS,m} - d * \left(w_{PMC-D1,DLS,m} - w_0\right)$$

Weight updates were determined by a learning rate ($\lambda$) which was specific to the striatal population and a decay rate ($d$). Weight decay was parameterized by the resting weight ($w_0$). If an update reduced a weight to be negative, it was reset to be 0.

## Behavioral task definitions

In a single trial, a simulation would produce the selection of an outcome and the selection of an action. These selections are based on the relative firing rates of the two populations found in each of the PFC and PMC. Outcome selection was determined by the population in the PFC with the highest firing rate. For example, if PFC population representing outcome #1 had a higher firing rate than the population representing outcome #2, it was said that outcome #1 was selected (see example in Fig 2C). In a similar fashion, action selection was determined by the PMC. If the PMC population representing action #1 had a higher firing rate than the population representing action #2, then it was said that action #1 was selected (see example in Fig 2C). Based on the selected action, some reward would be delivered, and this reward was used to update cortico-striatal weights. During the subsequent trial, the performance of action selection was determined by these newly updated weights. In this way, the model learns by incorporating reward feedback to alter the performance of action selection.

Our computational model was challenged with several behavioral tasks. We produced 100 instances of this model with different random number generator seeds, and each model instance was used to perform agent-based simulations. All agents were first trained with an initial learning session that was 200 trials long. During this session, the reward for each trial was set to 1 for the selection of action #1 and 0 for the selection of action #2. Following the initial learning session, each agent was challenged with five additional behavioral tasks which had different reward feedback rules: reward devaluation, reward reversal, punished outcome, reward reversal *PFC*[†], and punished outcome *PFC*[†] (see Table 4). In each of these sessions, the cortico-striatal weights were inherited from an initial learning session; in other words, agents were already trained to select action #1. In reward devaluation sessions, the reward for action #1 was set to 0.2. In reward reversal sessions, the reward was set to 0 for action #1 and 1 for action #2. In the punished outcome session, the reward was set to -0.5 for action #1 and 0 for action #2. The reward for the reward reversal *PFC*[†] and punished outcome *PFC*[†] sessions were identical to the reward reversal and punished outcome sessions, but the *PFC* neuronal

**Table 4. Behavioral sessions were defined by the organization of reward feedback.**

| Session Name | Action #1 feedback | Action #2 feedback | No. Trials |
|---|---|---|---|
| Initial Learning | 1 | 0 | 200 |
| Devaluation | 0.2 | 0 | 2000 |
| Reward reversal | 0 | 1 | 2000 |
| Punished outcome | -0.5 | 0 | 2000 |
| Reward reversal PFC[†] | 0 | 1 | 2000 |
| Punished outcome PFC[†] | -0.5 | 0 | 2000 |

population connectivity was altered in the $PFC^{†}$ sessions. Each of the five follow-up sessions contained 2000 trials.

To investigate the robustness of the model, we evaluated reward reversal and punished outcome sessions for some additional values of reward feedback (reward reversal: action #2 feedback at 0.2, 0.5, 1 and 2; punished outcome: action #1 feedback at -0.2, -0.5, -1, and -2).

## Evaluating model performance

The group performance of agents in reward reversal and punished outcome sessions was quantified using a normative model of evidence accumulation [126]. Change points in agent performance were measured by detecting zeros of the log likelihood ratio of outcome selection (based on PFC activity) and action selection (based on PMC activity). We defined the log likelihood ratio for cortical selection to be

$$y_n = \log\frac{P_1(w_n)}{P_2(w_n)} + \log\frac{(1-h)\exp(y_{n-1}) + h}{h\exp(y_{n-1}) + (1-h)}$$

The hazard coefficient $h$ was fixed to 1/201 as the reward feedback rules changed exactly once after the 200 trials of the Initial Learning session. When analyzing cortical activity, $P_1$ and $P_2$ were the likelihood of the agent selecting a particular outcome or action. This selection is determined by cortico-striatal weights ($w_n$ for the $n$th trial) as well as noise added to the initial conditions and in each step of numerical integration for that trial. We measured $P_1$ by caching the cortico-striatal weights for every trial in each session that an agent performs, repeating each trial 1000 times with different seeds for the pseudo random number generator, and recording the number of times outcome 1 was selected by the PFC or action #1 was selected by the PMC. We defined $P_2 = 1 - P_1$. The change point in PFC or PMC performance was determined by the trial number of the first zero-crossing of the log likelihood ratio $y$.

The difference in cortical selection performance of reward reversal and punished outcome sessions to reward reversal $PFC^{†}$ and punished outcome $PFC^{†}$ sessions was quantified by comparing the changepoints in PFC and PMC activity. Since changepoints indicated integer trial numbers, these comparisons were performed using the Mann-Whitney U test with an alpha level adjusted using the Bonferroni correction for four comparisons (0.05/4 or exactly 0.0125).

We evaluated performance of agents in the PFC and the PMC at the end of each behavioral session. Steady state performance of an individual agent was determined by $P_1$ and $P_2$ for the final trial of that session. We quantified the group difference between PFC and PMC performance using the Mann-Whitney U test with an alpha level determined by the Bonferroni correction for seven comparisons; the alpha value was 0.05 / 7 (approximately 7.1E-3). Only $P_1$ was used for these comparisons since $P_2$ was exactly determined by $P_1$.

## Software

Simulations were performed in Python using the JAX package [127]. Simulation results were analyzed using NumPy and SciPy [128, 129]. Statistics were computed using the wilcox.test function in the R computing environment. Figures were created with Matplotlib [130], Adobe Illustrator, and Microsoft PowerPoint. The source code used to perform these simulations is available at https://github.com/whbdupree/DMS_DLS_for_PLOSONE.

## Supporting information

**S1 Fig. An agent learns to select action #2 in a reward reversal behavioral task following impairment of executive function.** In panels depicting cortico-striatal weights, D1 synaptic weights are solid traces and D2 synaptic weights are dashed traces. Blue traces correspond to action #1 and orange traces correspond to action #2.
(TIF)

**S2 Fig. An agent learns to select action #2 in a punished outcome behavioral task following impairment of executive function.** In panels depicting cortico-striatal weights, D1 synaptic weights are solid traces and D2 synaptic weights are dashed traces. Blue traces correspond to action #1 and orange traces correspond to action #2.
(TIF)

**S3 Fig. We depict the evolution of corticostriatal weights for in the DMS and DLS for 100 model instances in initial learning, devaluation, reward and punished outcome sessions.** Sessions with different values for the reward magnitude are shown here for the reward and punished outcome challenges, and the value of reward is indicated in each subfigure title. Among the reward and punishment, the values used for simulations in the main results are indicated by an asterisk ($*$).
(PDF)

**S4 Fig. An agent has learned to select action #2 in a strongly (reward = -2) punished outcome behavioral task.** In panels depicting cortico-striatal weights, D1 synaptic weights are solid traces and D2 synaptic weights are dashed traces. Blue traces correspond to action #1 and orange traces correspond to action #2.
(TIF)

## Author Contributions

**Conceptualization:** William H. Barnett, Alexey Kuznetsov, Christopher C. Lapish.

**Data curation:** William H. Barnett.

**Formal analysis:** William H. Barnett.

**Funding acquisition:** Alexey Kuznetsov, Christopher C. Lapish.

**Investigation:** William H. Barnett.

**Methodology:** William H. Barnett, Alexey Kuznetsov, Christopher C. Lapish.

**Project administration:** Alexey Kuznetsov, Christopher C. Lapish.

**Resources:** William H. Barnett, Alexey Kuznetsov, Christopher C. Lapish.

**Software:** William H. Barnett.

**Supervision:** Alexey Kuznetsov, Christopher C. Lapish.

**Validation:** William H. Barnett, Alexey Kuznetsov, Christopher C. Lapish.

**Visualization:** William H. Barnett.

**Writing – original draft:** William H. Barnett, Alexey Kuznetsov, Christopher C. Lapish.

**Writing – review & editing:** William H. Barnett, Alexey Kuznetsov, Christopher C. Lapish.

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
