## [Decision Letter · Decision Letter 0]

12 Aug 2022

PONE-D-22-15624Distinct cortico-striatal compartments drive competition between adaptive and automatized behaviorPLOS ONE

Dear Dr. Barnett,

Thank you for submitting your manuscript to PLOS ONE. After careful consideration, we feel that it has merit but does not fully meet PLOS ONE’s publication criteria as it currently stands. Therefore, we invite you to submit a revised version of the manuscript that addresses the points raised during the review process. 

Following are the major points to be considered, based on my own and the reviewers’ reading of the paper:

- Clarity. At many places, it is hard to follow the model, the experimental paradigm, and how the model operates (see reviewer 1).  Some avenues to consider are a) including a figure with the basic experimental paradigm, so you can refer to it when you explain the model. b) Some high-level descriptions of how the model operates in the results section, in addition to the detailed explanation of which neurons are activated or not. The reader is asked to put in a lot of effort to follow the very busy pictures. Such pictures can be included, but there should also be pictures that show the main idea of how it operates.

Also, please explicitly consider the comment of reviewer 1 about the synaptic plasticity rule.

- Framing in the literature (reviewer 2). I agree with reviewer 2 that you should do a better job of comparing your work to earlier work. This also partially relates to the previous point; That you should give high-level pointers of how the model operates, differently than other models.

- Robustness. There are a lot of parameters in the model; it’s not clear (at least to me and reviewers) to what extent performance depends on such parameters. Relatedly, reviewer 2 asks to provide some indication of variability across simulations (so with a fixed parameter setting). You provide such indication for performance but not for weights. I’m not asking to re-draw all figures anew, but if we would have at least some idea of how variable neural responding is across trials with fixed parameters, that would be useful.

We look forward to receiving your revised manuscript.

Kind regards,

Tom Verguts

Academic Editor

PLOS ONE

Journal Requirements:

“This work was supported by P60-AA007611, AA029409, AA029970, and 5T32AA007462. This research was supported in part by Lilly Endowment, Inc., through its support for the Indiana University Pervasive Technology Institute.”

“5T32AA007462 to WHB from the National Institute on Alcohol Abuse and Alcoholism

https://www.niaaa.nih.gov

P60AA007611, AA029409, and AA029970 to CCL from the National Institute on Alcohol Abuse and Alcoholism

https://www.niaaa.nih.gov

This research was supported in part by Lilly Endowment, Inc., through its support for the Indiana University Pervasive Technology Institute.

Reviewers' comments:

Reviewer's Responses to Questions

**Comments to the Author**

1. Is the manuscript technically sound, and do the data support the conclusions?

Reviewer #1: Partly

Reviewer #2: Yes

2. Has the statistical analysis been performed appropriately and rigorously? 

Reviewer #1: Yes

Reviewer #2: Yes

3. Have the authors made all data underlying the findings in their manuscript fully available?

Reviewer #1: Yes

Reviewer #2: Yes

4. Is the manuscript presented in an intelligible fashion and written in standard English?

Reviewer #1: Yes

Reviewer #2: Yes

5. Review Comments to the Author

Reviewer #1: I have reviewed this paper for PLOS Computational Biology. I was disappointed that no changes were made considering the reviewers spent lots of time and provided many constructive suggestions. (Even typos were not corrected.) I strongly suggested the authors revised the manuscript carefully before submitting it to any journal.

Summary of the research and comments for the authors:

For decades, it has been proposed that basal ganglia and cortex are involved in goal-directed and habitual behaviors. However, there is no real consensus about how these circuits cooperate and modulate adaptive and automated behaviors. Barnett et al. developed a computational model to elucidate the underlying neural mechanisms. The model assumed that the dorsomedial striatum (DMS) received projections from the prefrontal cortex (PFC) and the dorsomedial striatum (DLS) received projections from the premotor cortex (PMC). PFC neurons further modulated the activities of neurons with the same action preference in the PMC. Actions were selected according to the PMC neurons’ responses. A value-based and a salience-based three-factor learning rule were adapted to update the DMS-PFC and DLS-PMC projections, respectively. As a result, PFC neurons encoded the desired outcome, and PMC neurons encoded the chosen action. The model was tested with reversal learning, devaluation, and punishment task and successfully reproduced animals’ adaptive behaviors. By misaligning PFC projections, the model mimicked the habitual responses that have been observed in the animals with degraded executive control.

In this manuscript, the distinct role of DMS and DLS is achieved by their different projections and learning rules which provides a new mechanism underlying adaptive behaviors and brings insight to the field. However, the results are not warranted to support their conclusions, and the new framework may lead to severe problems in certain circumstances. Also, many details need to be further clarified. I hope the following comments are constructive and help authors make a better manuscript.

Major Issues:

Network model

Some important details of the network model are missing; for example, what did excitatory drives look like? are they always constant? why did the STN only receive hyperdirect pathway input from the PMC, not PFC (Line 687-688)? Also, it was not represented in fig 1a. Would it change the results if both/neither PMC and/nor PFC project STN directly?

What was the ‘outcome’ (Line 138)? Was it the desired outcome/expected value of each action, or a binary variable indicating which action will lead to rewards? It should be clearly defined in the two-alternative forced-choice tasks used in the paper. And did the units in the PFC indeed encode the variables as you expected?

Synaptic plasticity

Line 234- 237. So if action 2 was chosen, outcome 1 neurons were activated and no reward was delivered based on the chosen action, then outcome # 1 neurons became less likely to be activated in the next trial. It doesn’t make sense. Do any studies support it? In contrast, many studies have shown that animals and humans can make counterfactual deductions and learn the value of the unchosen action correctly (Boorman et al. PLoS Biology 2011).

Only the connections between brain areas, instead of connections within CTX or DXS, were updated, indicating the information used for the outcome and action selection is stored in the long-range projections. Why not update the within-area connections? Do they lead to different results? The authors should at least discuss more whether the existing studies support it.

Line 279-280. ‘Since synaptic weights in the DMS were biased to promote both indirect pathways, the PFC was equally likely to select either outcome #1 or outcome #2 for several trials’. I am uncertain about this. Unless the weight promotes two pathways exactly equally, outcomes 1 and 2 won’t be chosen with the same probabilities.

Behaviors

The authors should mention how animals’ behaviors change after devaluation and punishment in the Results and cite relevant papers. Ideally, a comparison between animals’ and models’ behaviors should be made.

In the last section of the Results, the authors showed the steady performance of PFC and PMC, but talked nothing about whether these results, especially those from devaluation and punished outcome sessions, were supported by the experiments or not.

Punishment and salience/rectified expectation based learning rule

PMC-DLS system selects action based on its rectified expectation, which is defined as the absolute value of the expected reward with a temporal discounting. After choosing the punished action, the PMC-DLS D1 weight for the punished action would be enhanced and PMC-DLS D2 weight would decrease, which results in a higher probability of choosing the punished action. Even for the habitual system, a higher tendency to select the action after punishment is counterintuitive. Could you show how the choice history affects models’ decisions? I think you could see a trend of choosing the punished action when you use a larger magnitude of the punishment. If so, please explain.

Plot

I noticed that, in most panels, only the results from a single simulation were plotted. Of course, you can show that as an example, but it would be much better if you could show the averaged results over all simulations.

Minor Issues:

Line 89. Two periods.

The authors should use the word ‘exploration’ carefully. In the framework of learning theory and RL, exploration indicates that the agent/animal chooses the option with a lower expected value to obtain information. However, the authors used ‘exploration’ to describe behavior whenever the agent started selecting the other action after the reward contingency was changed, which was incorrect.

Line 571-572, since both D1 and D2 MSNs were involved in action selection, the prediction should be that both are critical for the transition to the exploratory behavior.

Line 749. ‘35utcome’ is a typo.

Line 728-731. According to the equations, the weight would not decay to the resting weight (w0). Please check the sign of the last term in these equations.

Figure 2. The title of the figure legend is missing.

Table 3, λ_(DLS,D2) was not in the list, and λ_(DMS,D2) should be positive. Please double-check these parameters.

Reviewer #2: The authors propose a computational model (built at the firing rate level of description) of two cortico-basal loops (a prefrontal and a premotor one) in order to study interactions of goal-directed and habitual behaviors in a number of classical conditioning tasks (reversal, devaluation, punishment), as well as under simulated executive control impairment.

* Previous computational models have proposed the cooperation of multiple learning systems with different properties to explain the cohabitation of (and the switches between) Goal-Directed (GD) vs. Habitual (H) behavior. The authors propose an interesting and relatively novel idea (that GD relies on model-free reinforcement learning and H on expected reward only), but completely fail to provide the current state-of-the-art, and to situate their proposal in the debate, which is problematic.

One of the first proposal in this domain is the (Daw et al., 2005) paper, that proposes that GD relies on model-based reinforcement learning (RL), and that H relies on model-free RL, the latter being slower to adapt than the former. This idea has been reused in many following studies (Dollé et al., 2010, Keramati et al., 2011, Pezzulo et al., 2013, etc.). Alternate view, with other algorithms mapped to GD and H, have also been proposed (see for example Dezfouli & Balleine, 2013, Topalidou et al., 2018, Geerts et al., 2020). The predictions that differentiate the behavior your new model from the existing ones, and could thus be used to tell them apart, are essential.

A proper state of the art is necessary, in the Introduction or in the Discussion of the present manuscript to compare with the previous proposals, highlight the differences, and how these differences allow for a better account of the phenomena of interest.

* Structuration of the model:

- section "Organization of the basal ganglia" (starting line 173):

the interpretation of the connectivity of the basal ganglia in terms of direct and indirect pathways was introduced by (Albin et al., 1989), a reference that is clearly missing here.

Moreover, this very simplistic interpretation of the BG circuitry has been augmented long time ago with the so-called hyper-direct pathway (Nambu et al., 2002), and the inclusion of the hyperdirect pathway is far from sufficient to really understand the circuit (Nambu 2008). Indeed, the interpretation in terms of pathways probably has to be reconsidered (Calabresi et al. 2014).

I understand that the authors won't build a new model of the basal ganglia and then perform again all their simulation, I am not asking for that. However, they cannot avoid mentioning this debate, and clearly state their position, either in this section, or in a dedicated part of their discussion.

- how is performed the final selection of an action? The methods adequately describe how the activity of the rate-coding neurons are computed and how learning is performed, however, how the activity in the simulated circuits is transformed in action 1 or action 2 being selected is unclear. Looking at figures 2-5, it seems that reward feedback is solely driven by the selection in the PMC, and that probably an activity threshold is used to determine whether the decision was made. Please describe the process precisely.

- many documented and functionally active connections are missing from the model (lines 676-680): no MSN-MSN inhibitions, no feedforward inhibition thanks to striatal interneurons, no feedback from the STN or the GPe to the MSNs, no projections from the GPe to the GPi. Why is that so, while many previous BG models have included them without difficulties? Moreover, the STN does not have the same connection pattern in the medial and the lateral circuit: why is the hyperdirect pathway excluded from PFC/Medial BG circuit? This pathway is thought to have a major contribution to the BG selection processes (Gurney et al., 2001a,b, Girard et al., 2021).

- Fig. 1A has to be corrected accordingly: the figure suggests that GPe projects to GPi, while it is not the case, according to the equations, and the PMC->STN projection is missing.

- the rationale for excluding the thalamus from the model and directly projecting the GPi outputs to the cortex also have to be justified.

- some explanations should also be provided with regards to the way the values of the parameters were chosen. Indeed, the simulation results will depend a lot on these parameters.

- the way it is currently formulated (line 723-731), the learning algorithm for the cortico-DLS weights is diverging. These weights continuously increase with expected reward. It does not appear as a desirable property for a biological system. Could the authors comment on this in the paper?

- line 335 and 752-753: what is the rationale for having a punishment value of 0.5, rather than the exact opposite of the initially provided reward (1 -> -1)?

* Results:

- In the reversal simulations, the agents manage to switch to an exploratory behavior in less than 20 trials, and to stabilize a new adapted behavior after 150 trials. How does it compare to animal data? This seems to be quite fast to be coined "habitual".

- Concerning reward devaluation, the authors write that their model suggests that the formation of habits is sensitive to the magnitude of the reinforcing stimulus. The line of work from Palminteri team, that started with (Palminteri et al., 2015), suggests than (in Humans at least), RL becomes relative with training (rather than absolute). Should it be the case, this sensitivity would disappear with training.

The same remark applies for the punishment simulations: with relative RL, a feedback of 0 acquires a positive value when the alternative choice has a negative value, and therefore the acquisition of an avoidance habit becomes possible (a possibility also suggested in LeDoux & Daw, 2018 and Geramita et al., 2020).

Therefore, these two predictions, specific to the proposed model, should be discussed in the context of these studies, for example around lines 501-502 and 514-516.

- The comparison of reversal versus punishment shows that the persistence in the less appropriate choice (option 1) is much longer in the punishment case. Can this specificity be compared to existing animal data?

- misalignment of PFC representation: could the authors comment the contents of fig. 6B and C? Specifically, the differences in PMC performance in the RR-PFC+ vs. the PO-PFC+ cases?

* Dicussion:

- could the authors be more specific, lines 571-572, when they write that D2 receptors are critical for the transition to exploratory behavior. To which parts of the results do they refer to?

- lines 621-625: what is the source of higher error rate in the PFC? To which extent does it depend on the parameterization of the model, or on its structure? I do not understand, in the light of the presented results, why "neuronal activity in the PFC should occasionally and increasingly correspond to alternative unrewarded actions as an animal learns to persistently select a rewarded action"? Why "increasingly"? I do not see what supports this "increase" in the results.

Minor remarks:

* The first subsections of the Results section ("Organization of cortico-striatal partitions", "Organization of the basal ganglia", "A biophysical model of the basal ganglia that includes both DMS and DLS") are not reporting results. They informally describe the model (while the formal description is provided in the final "Methods" section), and thus should be grouped together in another section, before the Results, describing the global structuration of the model.

* line 135: the organization of BG models in "choice-specific channels" is not an idea introduced by (Frank 2005), as it has been a standard in computational models since at least (Dominey & Arbib, 1992 ; Berns & Sejnowski, 1996 ; Gurney, Prescott, Redgrave, 2001a,b). The attribution has to be done correctly here.

* lines 191, 711 and 735: I do not agree with the use of "biophysical". The level of modeling used in this paper is not at the biophysical level, but phenomenological. Indeed, the parameters of table 3 cannot be directly related to physical dimensions.

* line 424: "box-and-whisker plots on Fig. 7B" isn't it Fig. 7C?

* Line 440-442: I have a problem with this sentence: the author proposed a model that used different dopaminergic signals in the two subdivisions of the striatum to propose an explanation of the GD and H learning, but I do not understand how one can "demontrate mechanisms" with such an approach.

* Line 542: in the performed simulations, what was impaired was not "executive control", but value attribution.

* line 719: waited -> weighted

* line 749: 35utcome -> outcome

References:

* Albin, Roger L., Anne B. Young, and John B. Penney. "The functional anatomy of basal ganglia disorders." Trends in neurosciences 12.10 (1989): 366-375.

* Berns, Gregory S., and Terrence J. Sejnowski. "How the basal ganglia make decisions." Neurobiology of decision-making. Springer, Berlin, Heidelberg, 1996. 101-113.

* Calabresi, P., Picconi, B., Tozzi, A., Ghiglieri, V., & Di Filippo, M. (2014). Direct and indirect pathways of basal ganglia: A critical reappraisal. Nature Neuroscience, 17(8), 1022–1030. https://doi. org/10.1038/nn.3743

* Daw, Nathaniel D., Yael Niv, and Peter Dayan. "Uncertainty-based competition between prefrontal and dorsolateral striatal systems for behavioral control." Nature neuroscience 8.12 (2005): 1704-1711.

* Dezfouli, Amir, and Bernard W. Balleine. "Actions, action sequences and habits: evidence that goal-directed and habitual action control are hierarchically organized." PLoS computational biology 9.12 (2013): e1003364.

* Dollé, Laurent, et al. "Path planning versus cue responding: a bio-inspired model of switching between navigation strategies." Biological cybernetics 103.4 (2010): 299-317.

* Dominey, Peter F., and Michael A. Arbib. "A cortico-subcortical model for generation of spatially accurate sequential saccades." Cerebral cortex 2.2 (1992): 153-175.

* Geerts, Jesse P., et al. "A general model of hippocampal and dorsal striatal learning and decision making." Proceedings of the National Academy of Sciences 117.49 (2020): 31427-31437.

* Geramita, Matthew A., Eric A. Yttri, and Susanne E. Ahmari. "The two‐step task, avoidance, and OCD." Journal of Neuroscience Research 98.6 (2020): 1007-1019.

* Girard, Benoît, et al. "A biologically constrained spiking neural network model of the primate basal ganglia with overlapping pathways exhibits action selection." European Journal of Neuroscience 53.7 (2021): 2254-2277.

* Gurney, Kevin, Tony J. Prescott, and Peter Redgrave. "A computational model of action selection in the basal ganglia. I. A new functional anatomy." Biological cybernetics 84.6 (2001): 401-410.

* Gurney, Kevin, Tony J. Prescott, and Peter Redgrave. "A computational model of action selection in the basal ganglia. II. Analysis and simulation of behaviour." Biological cybernetics 84.6 (2001): 411-423.

* Keramati, Mehdi, Amir Dezfouli, and Payam Piray. "Speed/accuracy trade-off between the habitual and the goal-directed processes." PLoS computational biology 7.5 (2011): e1002055.

* LeDoux, Joseph, and Nathaniel D. Daw. "Surviving threats: neural circuit and computational implications of a new taxonomy of defensive behaviour." Nature Reviews Neuroscience 19.5 (2018): 269-282.

* Nambu, Atsushi, Hironobu Tokuno, and Masahiko Takada. "Functional significance of the cortico–subthalamo–pallidal ‘hyperdirect’pathway." Neuroscience research 43.2 (2002): 111-117.

* Nambu, Atsushi. "Seven problems on the basal ganglia." Current opinion in neurobiology 18.6 (2008): 595-604.

* Palminteri, Stefano, et al. "Contextual modulation of value signals in reward and punishment learning." Nature communications 6.1 (2015): 1-14.

* Pezzulo, Giovanni, Francesco Rigoli, and Fabian Chersi. "The mixed instrumental controller: using value of information to combine habitual choice and mental simulation." Frontiers in psychology 4 (2013): 92.

* Topalidou, Meropi, et al. "A computational model of dual competition between the basal ganglia and the cortex." eneuro 5.6 (2018).

6. PLOS authors have the option to publish the peer review history of their article (what does this mean?). If published, this will include your full peer review and any attached files.

Reviewer #1: No

Reviewer #2: No

---

## [Author Response · Author response to Decision Letter 0]

5 Dec 2022

PONE-D-22-15624

Distinct cortico-striatal compartments drive competition between adaptive and automatized behavior

PLOS ONE

Dear Dr. Barnett,

Thank you for submitting your manuscript to PLOS ONE. After careful consideration, we feel that it has merit but does not fully meet PLOS ONE’s publication criteria as it currently stands. Therefore, we invite you to submit a revised version of the manuscript that addresses the points raised during the review process.

Following are the major points to be considered, based on my own and the reviewers’ reading of the paper:

- Clarity. At many places, it is hard to follow the model, the experimental paradigm, and how the model operates (see reviewer 1). Some avenues to consider are a) including a figure with the basic experimental paradigm, so you can refer to it when you explain the model. b) Some high-level descriptions of how the model operates in the results section, in addition to the detailed explanation of which neurons are activated or not. The reader is asked to put in a lot of effort to follow the very busy pictures. Such pictures can be included, but there should also be pictures that show the main idea of how it operates.

We have added an additional figure (new Figure 1) in order to describe the high level flow of our simulated behavioral experiments. We have also added a new supplemental figure (Figure S3) to show the evolution of corticostriatal synaptic weights at a high level as group data. It shows the evolution of all corticostriatal synaptic weights of 100 model instances for the duration of each simulated behavioral session. 

Also, please explicitly consider the comment of reviewer 1 about the synaptic plasticity rule.

It was not clear to us which comment of reviewer 1 is referenced, but we have attempted to thoroughly respond to all related comments by reviewer 1.

- Framing in the literature (reviewer 2). I agree with reviewer 2 that you should do a better job of comparing your work to earlier work. This also partially relates to the previous point; That you should give high-level pointers of how the model operates, differently than other models.

We have addressed this concern this by extensively modifying our review of related literature in the discussion (lines 572-585). We have also added relevant citations in the introduction to help frame the work (lines 108-110).

- Robustness. There are a lot of parameters in the model; it’s not clear (at least to me and reviewers) to what extent performance depends on such parameters. Relatedly, reviewer 2 asks to provide some indication of variability across simulations (so with a fixed parameter setting). You provide such indication for performance but not for weights. I’m not asking to re-draw all figures anew, but if we would have at least some idea of how variable neural responding is across trials with fixed parameters, that would be useful.

We have included an additional supplemental figure to address robustness (Figure S3). In this figure, we show the evolution of all corticostriatal synaptic weights of 100 model instances for the duration of each session. We also depict here simulated behavioral sessions for reward reversal and punished outcome not considered in the main text. In these additional results, we utilize values of reward feedback greater than or less than those presented in the main results. In this way, we show that our results are robust to modest variations in the magnitude of reward feedback.

We look forward to receiving your revised manuscript.

Kind regards,

Tom Verguts

Academic Editor

PLOS ONE

Journal Requirements:

“This work was supported by P60-AA007611, AA029409, AA029970, and 5T32AA007462. This research was supported in part by Lilly Endowment, Inc., through its support for the Indiana University Pervasive Technology Institute.”

“5T32AA007462 to WHB from the National Institute on Alcohol Abuse and Alcoholism

https://www.niaaa.nih.gov

P60AA007611, AA029409, and AA029970 to CCL from the National Institute on Alcohol Abuse and Alcoholism

https://www.niaaa.nih.gov

This research was supported in part by Lilly Endowment, Inc., through its support for the Indiana University Pervasive Technology Institute.

Reviewers' comments:

Reviewer's Responses to Questions

Comments to the Author

1. Is the manuscript technically sound, and do the data support the conclusions?

Reviewer #1: Partly

Reviewer #2: Yes

2. Has the statistical analysis been performed appropriately and rigorously?

Reviewer #1: Yes

Reviewer #2: Yes

3. Have the authors made all data underlying the findings in their manuscript fully available?

Reviewer #1: Yes

Reviewer #2: Yes

4. Is the manuscript presented in an intelligible fashion and written in standard English?

Reviewer #1: Yes

Reviewer #2: Yes

5. Review Comments to the Author

Reviewer #1: I have reviewed this paper for PLOS Computational Biology. I was disappointed that no changes were made considering the reviewers spent lots of time and provided many constructive suggestions. (Even typos were not corrected.) I strongly suggested the authors revised the manuscript carefully before submitting it to any journal.

We apologize that the reviewer received this submission a second time without revisions. We were instructed by journal staff to submit our manuscript without revisions in order to transfer to PLOS ONE:

“Dear Dr Barnett,

Thank you for your question. We are happy to facilitate a direct transfer to PLOS ONE, but would not be able to do so with a revised manuscript, as we are only able to transfer the manuscript version last submitted before the editorial decision was made. Please let us know if you would like to proceed with transferring the current version of the manuscript.

Kind regards,

Sarah

PLOS | OPEN FOR DISCOVERY

Sarah Mayo | Publishing Editor | Journal Editorial Operations

1160 Battery Street, Suite 225, San Francisco, CA 94111

smayo@plos.org | Main +1 415-624-1200 | Fax +1 415-546-4090”

Summary of the research and comments for the authors:

For decades, it has been proposed that basal ganglia and cortex are involved in goal-directed and habitual behaviors. However, there is no real consensus about how these circuits cooperate and modulate adaptive and automated behaviors. Barnett et al. developed a computational model to elucidate the underlying neural mechanisms. The model assumed that the dorsomedial striatum (DMS) received projections from the prefrontal cortex (PFC) and the dorsomedial striatum (DLS) received projections from the premotor cortex (PMC). PFC neurons further modulated the activities of neurons with the same action preference in the PMC. Actions were selected according to the PMC neurons’ responses. A value-based and a salience-based three-factor learning rule were adapted to update the DMS-PFC and DLS-PMC projections, respectively. As a result, PFC neurons encoded the desired outcome, and PMC neurons encoded the chosen action. The model was tested with reversal learning, devaluation, and punishment task and successfully reproduced animals’ adaptive behaviors. By misaligning PFC projections, the model mimicked the habitual responses that have been observed in the animals with degraded executive control.

In this manuscript, the distinct role of DMS and DLS is achieved by their different projections and learning rules which provides a new mechanism underlying adaptive behaviors and brings insight to the field. However, the results are not warranted to support their conclusions, and the new framework may lead to severe problems in certain circumstances. Also, many details need to be further clarified. I hope the following comments are constructive and help authors make a better manuscript.

Major Issues:

Network model

Some important details of the network model are missing; for example, what did excitatory drives look like? are they always constant? 

The excitatory drives (dr) were indeed always constant. Their values (dr_GPe, dr_STN, dr_GPi, dr_PFC) are each given in Table 3 of the Methods section (line 914).

why did the STN only receive hyperdirect pathway input from the PMC, not PFC (Line 687-688)? Also, it was not represented in fig 1a. Would it change the results if both/neither PMC and/nor PFC project STN directly?

We thank the reviewer for this comment, but we felt that extensive investigation involving the hyperdirect pathway to be outside the scope of this work. The architecture of this model was adapted from our previously published instances of this model (Kim et al., 2017; Mulcahy et al., 2020). We have added a passage about the hyperdirect pathway to the discussion section “Predictions and Limitations” to address the limitations of our implementation of the hyperdirect pathway and acknowledge other works that focus on this issue (lines 796-808). At present we do not simulate any behavioral tasks such as the stop-signal experiment that are used to specifically investigate these structures.

What was the ‘outcome’ (Line 138)? Was it the desired outcome/expected value of each action, or a binary variable indicating which action will lead to rewards? It should be clearly defined in the two-alternative forced-choice tasks used in the paper. And did the units in the PFC indeed encode the variables as you expected?

The outcome is a binary variable indicating which action will lead to a reward. We have updated text in the third paragraph of the “Organization of cortico-striatal partitions.” (lines 185-191) section of the results as well as the “Behavioral task definitions.” (lines 917-928) section of the Methods to clarify the outcome and action definitions.

Synaptic plasticity

Line 234- 237. So if action 2 was chosen, outcome 1 neurons were activated and no reward was delivered based on the chosen action, then outcome # 1 neurons became less likely to be activated in the next trial. It doesn’t make sense. Do any studies support it? In contrast, many studies have shown that animals and humans can make counterfactual deductions and learn the value of the unchosen action correctly (Boorman et al. PLoS Biology 2011).

Counterfactual deductions require model-based learning, and in the example provided above would require a mechanism to report the outcome of the non-selected action. Implementing this this capability via model-based learning is certainly an interesting future direction for our model. However, the present work does not implement a model-based learning controller, and we do not attempt to simulate a behavioral task that is designed to probe mechanisms for model-based learning such as the two-step task or the experiment described in Boorman et al. We have extended the second paragraph of the discussion (lines 572-585) to address previous work in goal-directed and habit-based learning that investigate model-based learning strategies that could make counterfactual deductions.

Only the connections between brain areas, instead of connections within CTX or DXS, were updated, indicating the information used for the outcome and action selection is stored in the long-range projections. Why not update the within-area connections? Do they lead to different results? The authors should at least discuss more whether the existing studies support it.

The architecture of this model was adapted from our previously published instances of this model (Kim et al., 2017; Mulcahy et al., 2020). These publications implemented and addressed plasticity within the cortex, but they did not implement plasticity of projections within the striatum. The presented architecture is sufficient to perform action selection tasks, and we consider plasticity within CTX or DXS to be outside the scope of the present publication. We have added text in the last paragraph of “The role and organization of direct and indirect pathways ” (lines 738-745) to acknowledge limitations in our implementation of the basal ganglia and contextualize our implementation within the scope of the study.

Line 279-280. ‘Since synaptic weights in the DMS were biased to promote both indirect pathways, the PFC was equally likely to select either outcome #1 or outcome #2 for several trials’. I am uncertain about this. Unless the weight promotes two pathways exactly equally, outcomes 1 and 2 won’t be chosen with the same probabilities.

We agree with the reviewer’s assessment. The inset figure 4B2 specifically calls out the trial in the session in which the weights for the two pathways become equal or close to being equal. We have added text to the results where figure 4B2 is referenced to clarify this issue (lines 340-3410.

Behaviors

The authors should mention how animals’ behaviors change after devaluation and punishment in the Results and cite relevant papers. Ideally, a comparison between animals’ and models’ behaviors should be made.

We have added text to the beginning of the devaluation (lines 367-377) and punishment (lines 418-426) results sections in order to better describe the simulated tasks. We have also added citations in both of those passages to contextualize the animal learning behavior. A passage was added to the discussion that addresses the comparison between model performance and animal behavior at the end of the “Reward devaluation and punishment reveal mechanisms that support habitual behavior” section (lines 661-675). Please continue reading on to our response to the next comment, which is intended to apply here, too.

In the last section of the Results, the authors showed the steady performance of PFC and PMC, but talked nothing about whether these results, especially those from devaluation and punished outcome sessions, were supported by the experiments or not.

A main goal of this study was to implement specific observations drawn from the literature (outlined in Table 1) and discuss the resulting emergent behavior of the model. We did not attempt to formally optimize model performance to fit animal behavior from some specific experiments. Moreover, it may not be possible to capture some aspects of animal behavior using only the regions of the brain represented in the present model. We have addressed this issue in a new passage added to the discussion (please see response to previous comment).

Punishment and salience/rectified expectation based learning rule

PMC-DLS system selects action based on its rectified expectation, which is defined as the absolute value of the expected reward with a temporal discounting. After choosing the punished action, the PMC-DLS D1 weight for the punished action would be enhanced and PMC-DLS D2 weight would decrease, which results in a higher probability of choosing the punished action. Even for the habitual system, a higher tendency to select the action after punishment is counterintuitive. Could you show how the choice history affects models’ decisions? I think you could see a trend of choosing the punished action when you use a larger magnitude of the punishment. If so, please explain.

We do observe transient “bad habits” when the punishment becomes large. Please see our text added to the end of the “Punishment engages goal-directed learning.” (lines 469-476) and the new supplemental figure which address this comment (Figure S4).

Plot

I noticed that, in most panels, only the results from a single simulation were plotted. Of course, you can show that as an example, but it would be much better if you could show the averaged results over all simulations.

We did attempt to capture average results in the group simulation and performance metrics that are depicted in Figures 2, 6, and 7. To address this comment and to demonstrate the robustness of our results, we have added a supplemental figure (S3) (first referenced lines 240-244) that depicts the evolution of cortico-striatal synaptic weights for 100 simulated agents across all sessions.

Minor Issues:

Line 89. Two periods.

This has been fixed.

The authors should use the word ‘exploration’ carefully. In the framework of learning theory and RL, exploration indicates that the agent/animal chooses the option with a lower expected value to obtain information. However, the authors used ‘exploration’ to describe behavior whenever the agent started selecting the other action after the reward contingency was changed, which was incorrect.

Thank you. We have added detail when we first use the word (in the Reward reversal engages goal directed learning to promote a new stimulus-response association. lines 345-347) in order to define in context how we are using the word exploration.

Line 571-572, since both D1 and D2 MSNs were involved in action selection, the prediction should be that both are critical for the transition to the exploratory behavior.

Thanks. We have clarified this prediction, which is found in the Discussion section “The role and organization of direct and indirect pathways” (line 723).

Line 749. ‘35utcome’ is a typo.

Thank you. This has been fixed.

Line 728-731. According to the equations, the weight would not decay to the resting weight (w0). Please check the sign of the last term in these equations.

Thank you. A typo in the decay term has been fixed (lines 906-909).

Figure 2. The title of the figure legend is missing.

We have added the missing figure legend. This is now Figure 3 (line 265).

Table 3, λ_(DLS,D2) was not in the list, and λ_(DMS,D2) should be positive. Please double-check these parameters.

We have corrected these errors (line 914).

Reviewer #2: The authors propose a computational model (built at the firing rate level of description) of two cortico-basal loops (a prefrontal and a premotor one) in order to study interactions of goal-directed and habitual behaviors in a number of classical conditioning tasks (reversal, devaluation, punishment), as well as under simulated executive control impairment.

* Previous computational models have proposed the cooperation of multiple learning systems with different properties to explain the cohabitation of (and the switches between) Goal-Directed (GD) vs. Habitual (H) behavior. The authors propose an interesting and relatively novel idea (that GD relies on model-free reinforcement learning and H on expected reward only), but completely fail to provide the current state-of-the-art, and to situate their proposal in the debate, which is problematic.

One of the first proposal in this domain is the (Daw et al., 2005) paper, that proposes that GD relies on model-based reinforcement learning (RL), and that H relies on model-free RL, the latter being slower to adapt than the former. This idea has been reused in many following studies (Dollé et al., 2010, Keramati et al., 2011, Pezzulo et al., 2013, etc.). Alternate view, with other algorithms mapped to GD and H, have also been proposed (see for example Dezfouli & Balleine, 2013, Topalidou et al., 2018, Geerts et al., 2020). The predictions that differentiate the behavior your new model from the existing ones, and could thus be used to tell them apart, are essential.

A proper state of the art is necessary, in the Introduction or in the Discussion of the present manuscript to compare with the previous proposals, highlight the differences, and how these differences allow for a better account of the phenomena of interest.

We agree that the current state-of-the-art is important to provide. The second paragraph of our Discussion specifically cites and extensively discusses the Habits Without Values paper by Miller et al., 2019 as well as Bogacz, 2020 and Baladron and Hamker, 2020, which we found to be the most recent publications directly applicable to the present work. We acknowledge that the text relies too heavily on the citation to Miller et al., 2019 to represent the body of preceding literature. To address the reviewer’s concerns, we extended the introduction and extensively rewrote this passage to better frame our discussion in the context of previous work in the reinforcement learning domain including most of the references mentioned above (lines 572-585). We have also included citations to this material in the Introduction (lines 108-110). 

* Structuration of the model:

- section "Organization of the basal ganglia" (starting line 173):

the interpretation of the connectivity of the basal ganglia in terms of direct and indirect pathways was introduced by (Albin et al., 1989), a reference that is clearly missing here.

Thank you calling this to our attention. We have added this important reference in the section “Organization of the basal ganglia” (line 210)

Moreover, this very simplistic interpretation of the BG circuitry has been augmented long time ago with the so-called hyper-direct pathway (Nambu et al., 2002), and the inclusion of the hyperdirect pathway is far from sufficient to really understand the circuit (Nambu 2008). Indeed, the interpretation in terms of pathways probably has to be reconsidered (Calabresi et al. 2014).

I understand that the authors won't build a new model of the basal ganglia and then perform again all their simulation, I am not asking for that. However, they cannot avoid mentioning this debate, and clearly state their position, either in this section, or in a dedicated part of their discussion.

We have addressed our implementation of the hyperdirect pathway in the “Predictions and limitations” section of the Discussion and included the two Nambu references. (lines 796-808)

We have extended the section “The role and organization of direct and indirect pathways” in the Discusssion to include reference to Calabresi et al. 2014. (lines 738-745)

- how is performed the final selection of an action? The methods adequately describe how the activity of the rate-coding neurons are computed and how learning is performed, however, how the activity in the simulated circuits is transformed in action 1 or action 2 being selected is unclear. Looking at figures 2-5, it seems that reward feedback is solely driven by the selection in the PMC, and that probably an activity threshold is used to determine whether the decision was made. Please describe the process precisely.

Thank you for this feedback. We have added some text to the methods to clarify how was action selection performed. This text can be found at the beginning of the “Behavioral task definitions.” (lines 917-924)

- many documented and functionally active connections are missing from the model (lines 676-680): no MSN-MSN inhibitions, no feedforward inhibition thanks to striatal interneurons, no feedback from the STN or the GPe to the MSNs, no projections from the GPe to the GPi. Why is that so, while many previous BG models have included them without difficulties? Moreover, the STN does not have the same connection pattern in the medial and the lateral circuit: why is the hyperdirect pathway excluded from PFC/Medial BG circuit? This pathway is thought to have a major contribution to the BG selection processes (Gurney et al., 2001a,b, Girard et al., 2021).

The architecture of this model was adapted from our previously published instances of this model (Kim et al., 2017; Mulcahy et al., 2020). These publications implemented and addressed plasticity within the cortex. The presented architecture was chosen as the minimum sufficient to perform action selection tasks, and we consider plasticity within CTX or DXS to be outside the scope of the present publication. We have added text in the last paragraph of “The role and organization of direct and indirect pathways ” to acknowledge limitations in our implementation of the basal ganglia and contextualize our implementation within the scope of the study (lines 738-745).

We have added a passage about the hyperdirect pathway to the discussion section “Predictions and Limitations” to address the limitations of our implementation of the hyperdirect pathway and acknowledge other works that focus on this issue. At present we do not simulate any behavioral tasks such as the stop-signal experiment that are used to specifically investigate these structures, and we felt that extensive investigation involving the hyperdirect pathway to be outside the scope of this work (lines 795-808).

- Fig. 1A has to be corrected accordingly: the figure suggests that GPe projects to Gpi, while it is not the case, according to the equations, and the PMC->STN projection is missing.

We have updated this figure. It is now Fig 2A.

- the rationale for excluding the thalamus from the model and directly projecting the GPi outputs to the cortex also have to be justified.

Cortico-thalamic dynamics are important for several phenomena, but their contribution to action selection is unclear. However, the thalamus is included in the current model albeit in a very limited manner. We do not attribute dynamics to thalamic neuronal populations, and we treat it as a relay from the output of the basal ganglia to the cortex. We have added text to the end of the “Predictions and Limitations” section to describe this limitation and included references to our previous publications from which this implementation was drawn. (Lines 823-827)

- some explanations should also be provided with regards to the way the values of the parameters were chosen. Indeed, the simulation results will depend a lot on these parameters.

The values of the biophysical parameters were drawn from the previous published instance of this model and tuned to accommodate the new model architecture and simulations that were performed here. Specifically, the adjustments were informed by the constraints that we describe in the Introduction (see Table 1). We have added text to the beginning of the Methods section to reflect this (lines 832-835)..

- the way it is currently formulated (line 723-731), the learning algorithm for the cortico-DLS weights is diverging. These weights continuously increase with expected reward. It does not appear as a desirable property for a biological system. Could the authors comment on this in the paper?

Thank you for drawing this to our attention. There was a typo involving the sign of the decay term in the manuscript, and it has been fixed (lines 906-909).

For the learning rate, reward values, and decay coefficient used in this work, the cortico-DLS weights saturate. We have visualized this observation in an additional supplemental figure (S3).

- line 335 and 752-753: what is the rationale for having a punishment value of 0.5, rather than the exact opposite of the initially provided reward (1 -> -1)?

We intended the punishment to not represent a stimulus that was injurious or too distracting, so we set the value to be relatively small. However, we have done additional simulations for punishment at different values to demonstrate the robustness of our results. Please see Figure S3. 

* Results:

- In the reversal simulations, the agents manage to switch to an exploratory behavior in less than 20 trials, and to stabilize a new adapted behavior after 150 trials. How does it compare to animal data? This seems to be quite fast to be coined “habitual”.

We have not attempted to fit the model to reproduce the timescale of acquisition of habit, and we consider the specific temporal properties of the witch to be outside the scope of the current study. Rather, we implement specific experimental observations and constraints (outline in Table 1 of the Introduction) and discuss the emergent results from simulated behavioral challenges. Also, the learning rates for the DMS and DLS could be scaled to accommodate specific experimental results.

- Concerning reward devaluation, the authors write that their model suggests that the formation of habits is sensitive to the magnitude of the reinforcing stimulus. The line of work from Palminteri team, that started with (Palminteri et al., 2015), suggests than (in Humans at least), RL becomes relative with training (rather than absolute). Should it be the case, this sensitivity would disappear with training.

The same remark applies for the punishment simulations: with relative RL, a feedback of 0 acquires a positive value when the alternative choice has a negative value, and therefore the acquisition of an avoidance habit becomes possible (a possibility also suggested in LeDoux & Daw, 2018 and Geramita et al., 2020).

Therefore, these two predictions, specific to the proposed model, should be discussed in the context of these studies, for example around lines 501-502 and 514-516.

Thank you for recommending the Palminteri et al., Ledoux & Daw, and Geramita et al. citations. We have extended the “Reward devaluation and punishment reveal mechanisms that support habitual behavior” section of the Discussion by adding a paragraph to address these issues (lines 661-675).

- The comparison of reversal versus punishment shows that the persistence in the less appropriate choice (option 1) is much longer in the punishment case. Can this specificity be compared to existing animal data?

It is difficult to compare the magnitude of a rewarding stimulus to an aversive stimulus. These characteristics must be tuned such that an animal is sufficiently motivated to seek reward or not sufficiently harmed or frightened to perform a task. As such, we have avoided a direct comparison between reversal and punished outcome sessions. Moreover, punishment learning engages regions of the brain that we have not incorporated in this model (for example the amygdala and the ventral striatum). We acknowledge this as a limitation, and we have incorporated a new passage to the Reward devaluation and punishment reveal mechanisms that support habitual behavior addressing some limitations in our implementation of punishment learning (lines 661-675).

- misalignment of PFC representation: could the authors comment the contents of fig. 6B and C? Specifically, the differences in PMC performance in the RR-PFC+ vs. the PO-PFC+ cases?

We have added text to the results in the section Misalignment of PFC outcome selection with channels in DMS and PMC decreases action selection performance that specifically call out and describe the results in what is now Fig. 7B and 7C (lines 497-502).

* Dicussion:

- could the authors be more specific, lines 571-572, when they write that D2 receptors are critical for the transition to exploratory behavior. To which parts of the results do they refer to?

We have added reference in this section of the discussion to the insets contained in what are now Fig 4B and Fig 6B that directly address these results (lines 718-719).

- lines 621-625: what is the source of higher error rate in the PFC? To which extent does it depend on the parameterization of the model, or on its structure? I do not understand, in the light of the presented results, why "neuronal activity in the PFC should occasionally and increasingly correspond to alternative unrewarded actions as an animal learns to persistently select a rewarded action"? Why "increasingly"? I do not see what supports this "increase" in the results.

This is a robust property revealed by the model in a wide range of parameters: As the rewarded action is persistently selected, the reward prediction error goes to zero. Thus, the learning signal that instructs the choice in the medial partition vanishes, and PFC increasingly samples the alternative outcome. We have added text to this passage in the Predictions and Limitations section of the discussion to clarify our prediction (lines 778-783).

Minor remarks:

* The first subsections of the Results section ("Organization of cortico-striatal partitions", "Organization of the basal ganglia", "A biophysical model of the basal ganglia that includes both DMS and DLS") are not reporting results. They informally describe the model (while the formal description is provided in the final "Methods" section), and thus should be grouped together in another section, before the Results, describing the global structuration of the model.

We have changed the section header to be Global Structure of the Model and moved the Results header to below these sections.

* line 135: the organization of BG models in "choice-specific channels" is not an idea introduced by (Frank 2005), as it has been a standard in computational models since at least (Dominey & Arbib, 1992 ; Berns & Sejnowski, 1996 ; Gurney, Prescott, Redgrave, 2001a,b). The attribution has to be done correctly here.

We have added these citations as directed (line 142).

* lines 191, 711 and 735: I do not agree with the use of "biophysical". The level of modeling used in this paper is not at the biophysical level, but phenomenological. Indeed, the parameters of table 3 cannot be directly related to physical dimensions.

We have altered the first instance to “firing-rate” and the second two instances to “model parameters” instead of “biophysical parameters” (lines 222, 889, 914).

* line 424: "box-and-whisker plots on Fig. 7B" isn't it Fig. 7C?

This has been fixed (line 555).

* Line 440-442: I have a problem with this sentence: the author proposed a model that used different dopaminergic signals in the two subdivisions of the striatum to propose an explanation of the GD and H learning, but I do not understand how one can "demontrate mechanisms" with such an approach.

We have changed this to “demonstrate computational mechanisms” (line 572)

* Line 542: in the performed simulations, what was impaired was not "executive control", but value attribution.

We understand the reviewers point. The difficulty with this interpretation is that the medial partition does not hold values. We have justified our interpretation and the definition of impaired executive control in the first two paragraphs of the results section Misalignment of PFC outcome selection with channels in DMS and PMC decreases action selection performance.

* line 719: waited -> weighted

Thanks. Fixed.

* line 749: 35utcome -> outcome

Thanks. Fixed.

References:

* Albin, Roger L., Anne B. Young, and John B. Penney. "The functional anatomy of basal ganglia disorders." Trends in neurosciences 12.10 (1989): 366-375.

* Berns, Gregory S., and Terrence J. Sejnowski. "How the basal ganglia make decisions." Neurobiology of decision-making. Springer, Berlin, Heidelberg, 1996. 101-113.

* Calabresi, P., Picconi, B., Tozzi, A., Ghiglieri, V., & Di Filippo, M. (2014). Direct and indirect pathways of basal ganglia: A critical reappraisal. Nature Neuroscience, 17(8), 1022–1030. https://doi. org/10.1038/nn.3743

* Daw, Nathaniel D., Yael Niv, and Peter Dayan. "Uncertainty-based competition between prefrontal and dorsolateral striatal systems for behavioral control." Nature neuroscience 8.12 (2005): 1704-1711.

* Dezfouli, Amir, and Bernard W. Balleine. "Actions, action sequences and habits: evidence that goal-directed and habitual action control are hierarchically organized." PLoS computational biology 9.12 (2013): e1003364.

* Dollé, Laurent, et al. "Path planning versus cue responding: a bio-inspired model of switching between navigation strategies." Biological cybernetics 103.4 (2010): 299-317.

* Dominey, Peter F., and Michael A. Arbib. "A cortico-subcortical model for generation of spatially accurate sequential saccades." Cerebral cortex 2.2 (1992): 153-175.

* Geerts, Jesse P., et al. "A general model of hippocampal and dorsal striatal learning and decision making." Proceedings of the National Academy of Sciences 117.49 (2020): 31427-31437.

* Geramita, Matthew A., Eric A. Yttri, and Susanne E. Ahmari. "The two‐step task, avoidance, and OCD." Journal of Neuroscience Research 98.6 (2020): 1007-1019.

* Girard, Benoît, et al. "A biologically constrained spiking neural network model of the primate basal ganglia with overlapping pathways exhibits action selection." European Journal of Neuroscience 53.7 (2021): 2254-2277.

* Gurney, Kevin, Tony J. Prescott, and Peter Redgrave. "A computational model of action selection in the basal ganglia. I. A new functional anatomy." Biological cybernetics 84.6 (2001): 401-410.

* Gurney, Kevin, Tony J. Prescott, and Peter Redgrave. "A computational model of action selection in the basal ganglia. II. Analysis and simulation of behaviour." Biological cybernetics 84.6 (2001): 411-423.

* Keramati, Mehdi, Amir Dezfouli, and Payam Piray. "Speed/accuracy trade-off between the habitual and the goal-directed processes." PLoS computational biology 7.5 (2011): e1002055.

* LeDoux, Joseph, and Nathaniel D. Daw. "Surviving threats: neural circuit and computational implications of a new taxonomy of defensive behaviour." Nature Reviews Neuroscience 19.5 (2018): 269-282.

* Nambu, Atsushi, Hironobu Tokuno, and Masahiko Takada. "Functional significance of the cortico–subthalamo–pallidal ‘hyperdirect’pathway." Neuroscience research 43.2 (2002): 111-117.

* Nambu, Atsushi. "Seven problems on the basal ganglia." Current opinion in neurobiology 18.6 (2008): 595-604.

* Palminteri, Stefano, et al. "Contextual modulation of value signals in reward and punishment learning." Nature communications 6.1 (2015): 1-14.

* Pezzulo, Giovanni, Francesco Rigoli, and Fabian Chersi. "The mixed instrumental controller: using value of information to combine habitual choice and mental simulation." Frontiers in psychology 4 (2013): 92.

* Topalidou, Meropi, et al. "A computational model of dual competition between the basal ganglia and the cortex." eneuro 5.6 (2018).

6. PLOS authors have the option to publish the peer review history of their article (what does this mean?). If published, this will include your full peer review and any attached files.

Do you want your identity to be public for this peer review? For information about this choice, including consent withdrawal, please see our Privacy Policy.

Reviewer #1: No

Reviewer #2: No

---

## [Editor Report · Decision Letter 1]

15 Dec 2022

Distinct cortico-striatal compartments drive competition between adaptive and automatized behavior

PONE-D-22-15624R1

Dear Dr. Barnett,

We’re pleased to inform you that your manuscript has been judged scientifically suitable for publication and will be formally accepted for publication once it meets all outstanding technical requirements.

Kind regards,

Tom Verguts

Academic Editor

PLOS ONE
---

## [Editor Report · Acceptance letter]

12 Mar 2023

PONE-D-22-15624R1 

Distinct cortico-striatal compartments drive competition between adaptive and automatized behavior 

Dear Dr. Barnett:

I'm pleased to inform you that your manuscript has been deemed suitable for publication in PLOS ONE. Congratulations! Your manuscript is now with our production department. 

Kind regards, 

on behalf of

Dr. Tom Verguts 

Academic Editor

PLOS ONE